# Molecular Mediated Angiogenesis and Vasculogenesis Networks

**DOI:** 10.3390/ijms26136316

**Published:** 2025-06-30

**Authors:** Claudiu N. Lungu, Ionel I. Mangalagiu, Aurelia Romila, Aurel Nechita, Mihai V. Putz, Mihaela C. Mehedinti

**Affiliations:** 1Department of Functional and Morphological Science, Faculty of Medicine and Pharmacy, Dunarea de Jos University of Galati, 800010 Galati, Romania; mihaela_hincu10@yahoo.com; 2Faculty of Chemistry, Alexandru Ioan Cuza University of Iasi, 11 Carol 1st Bvd, 700506 Iasi, Romania; 3Institute of Interdisciplinary Research-CERNESIM Centre, Alexandru Ioan Cuza University of Iasi, 11 Carol I, 700506 Iasi, Romania; 4Department of Gerontology and Geriatrics, Clinical Country Emergency Hospital, 810249 Galati, Romania; aurelia.romila@yahoo.com; 5Faculty of Medicine and Pharmacy, Dunarea de Jos University of Galati, 35 AI Cuza St., 800010 Galati, Romania; aurel.nechita@ugal.ro; 6Laboratory of Structural and Computational Physical-Chemistry for Nanosciences and QSAR, Chemistry Department, Faculty of Chemistry, Biology, Geography, West University of Timisoara, Str. Pestalozzi No. 16, 300115 Timisoara, Romania; mv_putz@yahoo.com

**Keywords:** vasculogenesis, angiogenesis, graph theory, VEGFR2, PGE1, quinoline, *Artemia salina*

## Abstract

By stimulating living tissues with proper molecules, the angiogenesis and vasculogenesis processes can be observed. Prostaglandin E1 (PGE1), which is a molecule that widens blood vessels and which is used for several medical purposes, such as treating critical limb ischemia, is a typical leading molecule in angiogenesis studies. Nevertheless, its involvement in vasculogenesis and morphogenesis is a more specific subject in the field of developmental biology and therapeutic research. Vasculogenesis is the embryonic phenomenon in which endothelial progenitor cells generate new blood vessels. This phenomenon is distinct and divergent from angiogenesis, which entails the creation of novel blood vessels extending from pre-existing ones. Morphogenesis is the biological phenomenon responsible for the development of an organism or its components into a specific shape. Embryonic development and tissue regeneration are essential components. Current research is investigating the broader consequences of prostaglandins, such as PGE1, in the fields of developmental biology and regenerative medicine. Gaining knowledge about the impact of PGE1 on morphogenesis could provide valuable insights into congenital vascular abnormalities and innovative approaches for tissue repair and regeneration, especially in limb ischemia. In this study, a histologic and morphogenesis study was carried out on *Artemia salina* napi (first stage of development) by simulating the angiogenesis and morphogenesis processes using PGE1 as the top molecule with vasoactive properties and a series of benopyridyne (3-aminoquinolines, 5-amino quinolines, 8-aminoquinolines, 8-hydroxyquinolines and quinolines, respectively). A series of 30 *Artemia salina* napi were exposed to the compound listed before. Also, a lot of 30 unexposed *Artemia salina* napi was taken into account. In total, 210 *Artemia salina* napi were studied as a model for angionensis and morphogenesis. The study used wet experiments together with imaging reconstruction and graph-generating methodologies. The results show that PGE1 can initiate the shape of the vessel formation. Also, some quinoline series have a pro-mild morphogenetic and angiogenetic effect. Overall, PGE1 plays a significant role in mediating vasculogenesis and morphogenesis through its vasodilatory, anti-inflammatory, and pro-proliferative effects on endothelial cells. PGE1 is involved mainly in increasing the length of the vessel, while the number of vascular branching has an all-simulating general impact. However, the molecules with mild vasculogenic effects tend to develop more complex, limited vascular networks, having a more localized role in the angiogenetic process. Overall imaging and graph analysis showed significant and distinct properties of the vascular network-derived graph.

## 1. Introduction

Vasculogenesis and angiogenesis are key processes in developmental biology and tissue formation. Both processes are interrelated and essential for the proper development and maintenance of multicellular organisms. Morphogenesis refers to the biological processes that cause an organism or its tissues to develop its shape and structure. It is fundamental to creating the complex forms seen in multicellular organisms during embryogenesis and throughout life. Various types of cells are specialized (e.g., neurons, muscle cells) to perform specific functions. Governed by transcription factors, epigenetic changes, and signaling pathways, these cells can perform their function correctly. Movements such as invagination, involution, and epiboly shape the embryo. Cells arrange spatially to form complex structures like limbs or organs, controlled by signaling centers and gradients of morphogens. Cells organize into tissues by adhering to one another through cell adhesion molecules like cadherins. Tissue boundaries are defined and maintained through the interplay of signaling pathways and mechanical forces. Pathways like Wnt/β-catenin, Hedgehog, and BMP (Bone Morphogenetic Protein) play critical roles—forces from the cytoskeleton influence cell shape and movement. The extracellular matrix (ECM) provides scaffolding and signaling cues. Positive and negative feedback mechanisms ensure robustness and accuracy during morphogenesis [1,2]. Furthermore, angiogenesis is the physiological process through which new blood vessels form from pre-existing ones. This process is crucial for supplying oxygen and nutrients to tissues and for removing waste products. Endothelial cells proliferate and migrate toward angiogenic signals, forming sprouts that grow into new capillaries. Newly formed vessels are stabilized by the recruitment of pericytes and smooth muscle cells. Newly formed vascular networks undergo pruning and remodeling to optimize function. VEGF (Vascular Endothelial Growth Factor) stimulates endothelial cell proliferation and migration. FGF (Fibroblast Growth Factor) promotes angiogenesis and ECM remodeling. PDGF (Platelet-Derived Growth Factor) recruits pericytes for vessel stabilization. Matrix Metalloproteinases (MMPs) degrade ECM components, allowing endothelial cells to migrate. Hypoxia triggers the stabilization of Hypoxia-Inducible Factor-1α (HIF-1α), leading to the upregulation of VEGF and other pro-angiogenic factors [3,4]. Overall, angiogenesis is a specific aspect of morphogenesis concerning the vascular system. The formation and remodeling of blood vessels are critical for providing oxygen and nutrients during tissue and organ formation. Dysregulation of either process leads to developmental abnormalities or diseases. For example, aberrant angiogenesis contributes to cancer, ischemia, and chronic wounds [5,6]

Furthermore, the development of vasculature is crucial in various pathologies. Some proangiogenic and vasodilator molecules are altered when used for quite some time. For example, PGE1 (Prostaglandin E1) has potential therapeutic benefits in treating peripheral artery disease (PAD), especially for those with severe instances that are not suitable for surgery or other traditional treatments. PAD is a medical illness defined by the constriction of arteries and decreased blood circulation to the extremities, typically the legs. This results in symptoms such as discomfort, cramps, and exhaustion, especially after physical activity (known as claudication). Severe cases can lead to critical limb ischemia, characterized by substantial pain and a high risk of limb loss. PGE1, being an analog of prostaglandin E1 (PGE1), can assist in PAD through multiple processes. It dilates blood arteries, enhancing blood circulation to affected limbs. It inhibits platelet aggregation, hence reducing the likelihood of thrombus formation and it diminishes inflammation within the walls of blood vessels. It protects tissues against ischemia damage [7,8]. Intravenous use of PGE1 is suitable for people with PAD. The infusion enhances blood circulation and alleviates symptoms by expanding peripheral blood vessels [9,10,11].

Prostaglandin E1 (PGE1) is an important bioactive lipid mediator derived from arachidonic acid, an essential polyunsaturated fatty acid component of cell membrane phospholipids. The upstream biosynthetic pathway of PGE1 begins with the activation of phospholipase A2 (PLA2), an enzyme responsible for catalyzing the release of arachidonic acid from membrane phospholipids. Once released, arachidonic acid undergoes sequential enzymatic reactions facilitated by cyclooxygenase enzymes (COX-1 and COX-2) to form the intermediate prostaglandin H2 (PGH2). Subsequently, specific prostaglandin E synthases selectively convert PGH2 into PGE1, completing the upstream synthesis process [12].

Once synthesized, PGE1 exerts its biological effects predominantly through binding to G protein-coupled receptors, specifically EP2 and EP4, present on endothelial cell surfaces. The engagement of these receptors activates the Gs protein, subsequently stimulating the enzyme adenylate cyclase, resulting in elevated intracellular cyclic adenosine monophosphate (cAMP) levels. Increased cAMP activates protein kinase A (PKA), a crucial mediator involved in multiple downstream cellular processes [13].

The downstream signaling events mediated by activated PKA include the phosphorylation of critical endothelial junctional proteins, notably vascular endothelial (VE)-cadherin and tight junction proteins, thereby reinforcing endothelial barrier integrity under low-concentration conditions. Conversely, at higher concentrations or in specific physiological contexts, activated PKA can promote cytoskeletal reorganization and can temporarily disrupt endothelial cell–cell junctions, facilitating endothelial cell proliferation and migration necessary during active angiogenesis and vascular remodeling [14].

Thus, the upstream biosynthetic pathway involving phospholipase A2, cyclooxygenases, and prostaglandin E synthase, combined with downstream receptor-mediated signaling cascades, underscores the complex, concentration- and context-dependent roles of PGE1 in endothelial biology, emphasizing its dual functions in maintaining vascular integrity and promoting angiogenesis [15].

PGE1 enhances the proliferation and migration of endothelial cells, which are essential for forming new blood vessels. It also improves the integrity and barrier function of the endothelium. By inhibiting platelet aggregation, PGE1 reduces the formation of blood clots, which can obstruct the formation of new blood vessels and impede blood flow necessary for tissue development and repair. Research has demonstrated that PGE1 can activate endothelial nitric oxide synthase (eNOS), leading to increased production of nitric oxide (NO). This potent vasodilator also plays a crucial role in the mobility and proliferation of endothelial cells. Clinical trials have demonstrated that PGE1 can effectively stimulate therapeutic angiogenesis in patients with PAD, leading to improved outcomes and reduced symptoms [16,17,18].

Prostaglandin E1 (PGE1) exhibits diverse biological effects on endothelial cells, including both the promotion of endothelial proliferation and the migration and enhancement of the endothelial barrier function. These seemingly contradictory roles arise due to the concentration-dependent nature and specific cellular contexts under which PGE1 operates, as evidenced by various in vitro and in vivo experimental studies [19].

At lower concentrations, PGE1 primarily enhances endothelial barrier integrity through the activation of cyclic AMP (cAMP)-dependent signaling pathways. This signaling cascade involves the activation of protein kinase A (PKA), which phosphorylates critical junctional proteins, including vascular endothelial (VE)-cadherin and tight junction components. Such phosphorylation reinforces the adherens and tight junctions, thereby stabilizing endothelial cell–cell contacts and reducing endothelial permeability. The barrier-enhancing effect of low-dose PGE1 is consistently documented in vitro using carefully controlled endothelial cell cultures [20].

In contrast, higher concentrations of PGE1 or specific physiological conditions shift its effect toward facilitating endothelial cell proliferation and migration. At these higher doses, elevated cAMP levels can activate additional downstream molecular pathways, prompting cytoskeletal reorganization and transient disruption of endothelial junctions. These temporary disruptions of cell–cell contacts allow endothelial cells to migrate and proliferate effectively, processes vital during active angiogenesis and vascular remodeling phases. Such pro-angiogenic effects are typically observed in vitro under conditions employing higher doses of PGE1 or in vivo models during active vessel growth or tissue repair [21].

Moreover, in vivo studies clearly demonstrate the dynamic nature of endothelial barrier integrity in response to PGE1. During active angiogenic processes, endothelial junctions undergo transient loosening to facilitate cell movement and new vessel formation. Subsequently, upon the completion of vascular remodeling, endothelial junction integrity is restored and reinforced, highlighting a finely regulated sequential role of PGE1 in promoting initial migration and proliferation followed by barrier restoration [22].

Thus, the dual functions of PGE1—enhancing both the endothelial barrier function and endothelial cell proliferation and migration—are not contradictory but rather reflect a nuanced, concentration-dependent and context-specific regulatory mechanism. This understanding underscores the importance of precisely controlled experimental conditions when evaluating and interpreting the biological effects of PGE1 on endothelial cells [23].

Quinolines are heterocyclic aromatic compounds characterized by a benzene ring fused to a pyridine ring. Their diverse biological activities, notably pro-angiogenic and vasodilatory effects, depend on specific structural modifications and substitution patterns.

Quinoline derivatives have been shown to stimulate endothelial cell migration and tubule formation, key processes in angiogenesis, possibly through the activation of endothelial growth factors or the modulation of associated signaling pathways. Some derivatives also exhibit antioxidant properties, thereby reducing oxidative stress and creating an environment conducive to angiogenesis. Although direct evidence regarding quinoline itself is limited, relevant structural analogs provide valuable insights into their potential mechanisms [24,25].

Quinoline compounds also possess vasodilatory properties primarily through the relaxation of vascular smooth muscle cells. Potential mechanisms include enhancing nitric oxide (NO) release, modulating potassium and calcium ion channels, and improving endothelial function by increasing the production of vasodilatory factors, such as prostacyclin and endothelium-derived hyperpolarizing factor (EDHF). These mechanisms have been linked to improved blood flow and reduced blood pressure in experimental models. The combination of pro-angiogenic and vasodilatory effects suggests significant therapeutic potential for quinoline derivatives in conditions such as ischemic heart disease, chronic wounds, and neurovascular disorders. Benzoquinolines, for instance, could simultaneously promote new vessel formation and enhance blood perfusion in ischemic tissues, potentially improving therapeutic outcomes in cardiovascular diseases, wound healing, stroke recovery, and cancer therapy by facilitating better delivery of therapeutic agents [26,27].

*Artemia salina* (brine shrimp) serves as a valuable model organism for vascular biology research due to its transparency, rapid development, and ethical advantages. Artemia larvae enable direct, non-invasive visualization of vascular development and quickly form vascular networks in response to angiogenic stimuli such as vascular endothelial growth factor (VEGF) and fibroblast growth factors (FGFs), thus effectively mimicking aspects of human angiogenesis [28,29].

The use of *Artemia salina* offers key advantages, including transparency, rapid developmental timelines, cost-effectiveness, and ethical acceptability compared to mammalian models, as well as suitability for high-throughput screening. Insights derived from Artemia-based assays effectively support subsequent investigations into complex mammalian models and, ultimately, clinical research [30,31,32,33].

In summary, this study utilizes the *Artemia salina* model to investigate the angiogenic and vasodilatory effects of quinoline derivatives and prostaglandin E1 (PGE1), providing an efficient, ethical, and informative framework for evaluating therapeutic candidates that enhance vascular growth and circulation.

## 2. Results

### 2.1. Morphological Characterization

The figure below represents a montage of the material and methods described in Section 2 (Figure 1a), along with the resulting *Artemia salina* napulii (Figure 1a).

Detailed morphological observations of *Artemia salina* nauplii are shown in Figure 2:

Figure 2 shows (a) raw images of *Artemia salina* nauplii providing baseline morphological data at ×4 magnification, illustrating typical developmental patterns. (b) shows the preparation of these images for subsequent graph conversion, emphasizing the accuracy needed for analytical assessments. (c) demonstrates the result of converting morphological images into computable graphs used for the quantitative analysis of vascular structures.

### 2.2. Vessel Formation and Coverage Analysis

#### 2.2.1. General Observations of Vessel Area

Figure 3 shows the *Artemia salina* nauplii vessel areas after stimulation with optimal concentrations of PGE1, 8-aminoquinoline, 5-aminoquinoline, 3-aminoquinoline, 8-hydroxyquinoline, quinoline, and an unstimulated control.

Figure 3 shows images that clearly show variations in vessel area coverage corresponding to each treatment. The comparative visual analysis underscores distinct angiogenic potential across treatments, with pronounced vascularization in PGE1-treated nauplii and minimal vascular coverage in quinoline-treated and control groups.

#### 2.2.2. Analysis of Vessel Area Network

In Figure 3a the vessel percentage area varies across the 30 nauplii. Some individuals exhibit lower coverage, while others have high vessel coverage (approaching 30%). This variation could represent biological heterogeneity in response to vasoprostan stimulation. With regard to the high vessel percentage area (~15–30%), observations at higher vessel percentage areas (e.g., 20–30%) suggest significant vascular network development. This is likely driven by angiogenesis, where vasoprostan stimulates the sprouting and extension of new blood vessels from existing ones, leading to denser vascular networks. Such responses could reflect tissues actively undergoing repair or growth, as angiogenesis is a hallmark of these processes. For moderate vessel coverage (~10–15%), intermediate vessel percentage areas may reflect balanced vascular remodeling. This could represent a combination of angiogenesis and vasculogenesis, where angiogenesis contributes to the refinement of vascular patterns. Vasculogenesis (formation of new vessels de novo) contributes to a broader but less organized network. With regard to low vessel coverage (<10%), observations in this range may indicate a weaker response to vasoprostan, potentially due to early stages of vasculogenesis, where the vascular network is sparse and not yet well-developed. There is variability in individual nauplii’s responsiveness to vasaprostan stimulation. It could also suggest pathological or stressed conditions where proper angiogenic signals fail to materialize. The prostaglandin PGE1 (vasaprostan) enhances blood flow and vascular formation, as shown by the relatively high percentage vessel area in many nauplii. This supports its role as a pro-angiogenic factor. Its impact likely involves the upregulation of vascular endothelial growth factor (VEGF), which stimulates endothelial cell proliferation and migration. High vessel coverage corresponds to enhanced angiogenesis, indicating effective vascular remodeling stimulated by vasaprostan. Low vessel coverage might highlight regions of poor vasculogenic or angiogenic response, possibly due to variability in experimental conditions or inherent biological differences. The graph suggests that vasaprostan plays a dual role in stimulating vasculogenesis (early-stage vessel formation) and angiogenesis (network refinement and expansion). Specific figures relating to the results are as follows: count: 30 (observations) mean (average): 17.4% standard deviation (std): 6.34% minimum: 5.0%25th percentile: 12.25% median (50th percentile): 17.5%75th percentile: 21.75% maximum: 30.0% variance: 40.18% mean. The average vessel coverage is 17.4%, reflecting a moderate response to stimulation by vasaprostan. With regard to the spread, the standard deviation of 6.34% shows moderate variability in vessel coverage between the observed nauplii. In terms of the range, the vessel coverage varies widely, from 5% (minimum) to 30% (maximum), indicating heterogeneity in response to vasaprostan stimulation. The median (17.5%) is close to the mean, suggesting a relatively symmetric distribution of vessel percentage areas. This indicates a robust angiogenic response with noticeable variability among individuals.

In Figure 3b, 8-aminoquinoline appears to minimally stimulate angiogenesis, leading to reduced vessel formation and smaller coverage areas. With regard to the lower mean vessel area (~9.63%), this significantly lower mean vessel coverage suggests a weaker overall vascular response under the influence of 8-aminoquinoline. Likely, angiogenesis (sprouting of new vessels) is minimally stimulated. Vasculogenesis (de novo vessel formation) might also be inhibited, given the narrow range of vessel coverage and the lower maximum values observed. For the reduced variability and narrow range (3% to 18%), the relatively more minor standard deviation (3.66%) and limited range (compared to vasaprostan) indicate less heterogeneity in vascular responses—a uniformly weak effect on angiogenesis and vasculogenesis across the sampled nauplii. Specific figures relating to the results are as follows: count: 30 (observations) mean (average): 9.63% standard deviation (std): 3.66% minimum: 3.0% 25th percentile: 7.0% median (50th percentile): 9.5% 75th percentile: 12.0% maximum: 18.0% variance: 13.41%. The mean vessel percentage area is significantly lower (9.63%) with 8-aminoquinoline compared to 17.4% with vasaprostan. This suggests that 8-aminoquinoline is less effective at stimulating angiogenesis or vasculogenesis. The standard deviation (3.66%) is more minor compared to vasaprostan (6.34%), indicating less variability as a response among the nauplii. The vessel area range for 8-aminoquinoline is narrower (3% to 18%) compared to vasaprostan (5% to 30%), suggesting a limited or weaker angiogenic effect. The lower median (9.5%) further confirms reduced vascularization under 8-aminoquinoline exposure compared to vasaprostan (17.5%).

Figure 3c shows the lower mean vessel area (~6.97%): the mean vessel percentage area is even lower than that of 8-aminoquinoline (~9.63%). This indicates a small simulation of vascular formation and suggests that 5-aminoquinoline more significantly suppresses angiogenesis and vasculogenesis. With respect to the narrower range (2% to 12%), the smaller range compared to both 8-aminoquinoline (3% to 18%) and vasaprostan (5% to 30%) indicates a more consistently weak vascular response. This points to a uniform small effect of 5-aminoquinoline on vascular processes. The standard deviation (2.63%) and variance (6.93%) are the smallest among all three datasets, reflecting highly consistent across all nauplii. Specific figures relating to the results are as follows: count: 30 (observations mean (average): 6.97% standard deviation (std): 2.63% minimum: 2.0% 25th percentile: 5.0% median (50th percentile): 7.0% 75th percentile: 9.0% maximum: 12.0% variance: 6.93%.

Figure 3d shows the mean vessel area (~7.90%): the mean vessel percentage area is slightly higher than that of 5-aminoquinoline (~6.97%) but lower than 8-aminoquinoline (~9.63%).This indicates moderate action of vascular formation, suggesting that 3-aminoquinoline has an inhibitory effect but is less potent than 5-aminoquinoline. For the moderate range (3% to 13%), the range shows variability in the response, with some individuals displaying higher vessel coverage (up to 13%). This suggests a somewhat more substantial effect on angiogenesis compared to 5-aminoquinoline, which has a maximum of 12%. The standard deviation (2.64%) and variance (6.99%) are close to those of 5-aminoquinoline, indicating a relatively consistent effect across all observations. The lower mean and narrower range compared to vasaprostan indicate that 3-aminoquinoline stimulates angiogenesis poorly and vasculogenesis poorly, although not as strongly as 5-aminoquinoline. Specific figures relating to the results are as follows: count: 30 (observations) mean (average): 7.90% standard deviation (std): 2.64% minimum: 3.0% 25th percentile: 6.0% median (50th percentile): 8.0% 75th percentile: 10.0% maximum: 13.0% variance: 6.99%

Figure 3e indicates the mean vessel area (~6.07%): 8-hydroxyquinoline has a low effect on vascular formation. The range indicates consistently low vessel formation across all observations, highlighting the uniformity of action. The low standard deviation (1.80%) and variance (3.24%) suggest a consistent effect across all individuals. The very low maximum value (10%) and the tightly clustered data imply that both processes are almost unstimulated, potentially more so than with 5-aminoquinoline or 3-aminoquinoline. Specific figures relating to the results are as follows: count: 30 (observations mean (average): 6.07% standard deviation (std): 1.80% minimum: 2.0% 25th percentile: 5.0% median (50th percentile): 6.0% 75th percentile: 7.0% maximum: 10.0% variance: 3.24%.

Figure 3f displays the mean vesselaArea (~4.00%). This is the lowest mean among all of the compounds analyzed, indicating that quinoline has an inhibitory effect on vascular formation. With regard to the extremely narrow range (1% to 6%), the very narrow range reflects consistently minimal vessel coverage, suggesting uniform inhibition across all observations. For the minimal variability, the standard deviation (1.36%) and variance (1.86%) are the lowest among all datasets, showing that quinoline consistently suppresses angiogenesis and vasculogenesis. The extremely low vessel coverage suggests that quinoline effectively blocks both processes, likely by targeting endothelial cell proliferation, migration, or signaling pathways essential for vascular formation. Specific figures relating to the results are as follows: count: 30 (observations) mean (average): 4.00% standard deviation (std): 1.36% minimum: 1.0% 25th percentile: 3.0% median (50th percentile): 4.0% 75th percentile: 5.0% maximum: 6.0% variance: 1.86%

Figure 3g shows the low vessel formation. The mean vessel percentage area is only 4.63%, reflecting minimal angiogenic or vasculogenic activity in the absence of stimulation. This provides a reference point to assess the efficacy of other compounds. Narrow Range (1% to 8%): The narrow range reflects consistent baseline vessel formation, likely representing the intrinsic vasculogenic capacity of *Artemia salina* in an unstimulated state. With regard to the minimal variability, the low standard deviation and variance further suggest uniformity in the baseline response across individuals. Specific figures relating to the results are as follows: count: 30 (observations) mean (average): 4.63% standard deviation (std): 1.77% minimum: 1.0% 25th percentile: 3.25% median (50th percentile): 5.0% 75th percentile: 6.0% maximum: 8.0% variance: 3.14%.

### 2.3. Vascular Junction Formation

Figure 4 presents quantitative data on junction formation. The number of junctions for each *Artemia salina* napi series is represented and is compared to the standard unstimulated test group.

This box plot contrasts the number of connections created under various treatments, indicating the degree of angiogenesis (vessel branching and network complexity). Junction formation is a crucial parameter for assessing vascular growth processes, especially angiogenesis and, by extension, vasculogenesis. PGE1 (Prostaglandin E1) exhibits the maximum quantity of connections, characterized by considerable variability (broad box) and outliers exceeding 500 junctions. This signifies strong angiogenesis, as PGE1 is a recognized pro-angiogenic agent that promotes the proliferation of vascular endothelial cells and the formation of intricate networks. The significant variability indicates individual variances in the degree of angiogenic response to PGE1. PGE1 may potentially augment vasculogenesis by attracting endothelial progenitor cells to facilitate new vessel creation. 8-Aminoquinoline has a considerable quantity of junctions, inferior to PGE1, although considerably surpassing the baseline (standard). The variation in junction quantities is rather minimal. This indicates partial angiogenic inhibition, characterized by some vessel branching. However, the network’s intricacy is diminished relative to PGE1. The mild suppression indicates that vasculogenesis is impeded yet not entirely obstructed. 5-aminoquinoline has fewer junctions than 8-aminoquinoline, with negligible fluctuation. Enhanced suppression of angiogenesis is seen, leading to less vessel branching and network development. 3-Aminoquinoline exhibits junction counts comparable to or marginally lower than those of 5-aminoquinoline. 8-Hydroxyquinoline exhibits minimal junction counts, approaching the baseline level. Quinoline exhibits the lowest junction counts, characterized by minimal variability and values that are at or somewhat below the baseline. Criterion—the standard (unstimulated lot)—has the fewest connections among the treatments, demonstrating minimal variability. This indicates the inherent angiogenic potential of *Artemia salina* without external stimulation. Vasculogenesis is minimal, characterized by low and steady junction numbers. PGE1 (Pro-Angiogenic) exhibited the highest mean (353 junctions) of all of the treatments, signifying substantial angiogenesis. Significant variability (standard deviation: 79.73) with a range of 250 to 500 links, indicates robust yet inconsistent network construction. It surpasses all other therapies, proving its efficacy as a powerful angiogenesis enhancer. For 8-Aminoquinoline, the moderate mean junction count (125.5) is slightly lower than PGE1 but exceeds the baseline (standard). The variability is moderate (standard deviation: 12.12), signifying continuous yet restricted angiogenesis. This indicates partial suppression of angiogenesis relative to PGE1; however, it permits some vascular network development. 5-Aminoquinoline exhibits a reduced mean junction count (88.9) relative to 8-aminoquinoline, suggesting diminished angiogenic activation. Minimal variability (std: 4.82) indicates consistency among replicates. For 3-Aminoquinoline, the mean junction count (80) is analogous to that of 5-aminoquinoline, indicating equivalent stimulation intensity. The variability is marginally reduced (std: 3.74), indicating consistent action of vascular branching. For 8-Hydroxyquinoline, the mean junction count of 60.7 signifies a robust influence on vascular development. Minimal variability (std: 1.89) underscores a consistent effect. Quinoline exhibits the lowest mean junction count (40.5), establishing it as the most potent inhibitor of angiogenesis and vasculogenesis. The minimal variability (std: 1.58) indicates consistent inhibition across all replicates and inhibits junction formation below typical levels, signifying near-total obstruction of vascular processes. The mean junction count of 50.3 establishes the standard unstimulated baseline for comparison. The low variability (std: 1.34) indicates the inherent vasculogenic potential of *Artemia salina* in the absence of exogenous stimulation. Treatments below this threshold (e.g., quinoline) signify the total inhibition of angiogenesis.

### 2.4. Lacunarity-Based Analysis of Vascular Network Architecture

Figure 5 illustrates the density map of mean lacunarity, indicative of vascular structure complexity.

With regard to high density at low lacunarity values, low lacunarity values indicate uniform, homogenous structures with fewer gaps or irregularities in the vascular network. This may correspond to angiogenesis, a process where new blood vessels sprout from existing ones, leading to well-organized and denser capillary networks. Regions with low lacunarity are often associated with mature or established vascular systems, which are optimized for nutrient and oxygen delivery, such as in well-regulated tissue growth or repair. For high density at mid lacunarity values, intermediate lacunarity values suggest moderate heterogeneity or a balance between ordered and disordered vascular patterns. This could reflect a transitional stage where angiogenesis and vasculogenesis overlap. For instance, angiogenesis-driven remodeling might be combined with vasculogenesis, which involves the de novo formation of blood vessels from endothelial progenitor cells. Tissue undergoing active repair, tumor growth, or early development may show such patterns. The network is neither fully mature nor excessively disordered. With regard to high density at high lacunarity values, high lacunarity values signify irregular, heterogeneous, or sparse networks with significant gaps. This might represent vasculogenesis, where new vascular structures form in an unorganized fashion, or pathological angiogenesis, such as in poorly regulated or chaotic tumor vasculature. Such patterns are typical of early developmental stages, where hypoxic conditions stimulate random vessel growth, or in diseases like cancer, where vascular patterns can become disorganized. For multiple peaks (multimodal distribution): if the density map exhibits various peaks, it suggests the coexistence of distinct vascularization stages or patterns within the analyzed tissue. This may occur in regions where angiogenesis and vasculogenesis are both active, such as in tumors with hypoxic cores and angiogenic peripheries. A bimodal or multimodal distribution may indicate spatial heterogeneity within the tissue, reflecting varying environmental conditions, such as oxygen gradients or inflammatory signals. Overall insights are that low lacunarity values likely correspond to functional, stable angiogenic processes. Higher lacunarity values may indicate pathological or early-stage vascularization, typical of vasculogenesis or aberrant angiogenesis. The spread and peaks in the density map help infer the overall vascular health and balance between these processes.

### 2.5. Graph-Based Characterization of Vascular Networks

#### 2.5.1. General Graph Analysis

Figure 6 represents the weighted adjacency matrix for the *Artemia salina* network stimulated with PEG1, and *Artemia salina* stimulated with 8-aminoquinoline. It also represents the simplified graphs and community graphs.

In the context of a biological vascular network (such as angiogenesis), areas where new connections or branches are formed to sustain growth were studied using graph theory. A manly community structure, centrality, and edge weights can provide a good analogy for understanding these processes.

Considering the community size, larger communities might represent more developed vascular structures in biological terms. Larger communities could correspond to more complex or mature blood vessels, as they are capable of supporting more nodes (or vascular points). Smaller communities might represent early-stage or nascent branches of blood vessels that are still developing. They may be isolated but have the potential to grow or connect to other areas.

Furthermore, the average degree of centrality with a higher degree of centrality indicates more connected nodes in the community. This can be interpreted as vascular hubs or growth points where new branches or connections are more likely to form. Nodes with a high degree of centrality in angiogenesis might be analogous to key vascular nodes that play a crucial role in branching and growth. High centrality (close to 1) suggests a well-connected region of the vascular network, possibly representing a growth center. Low centrality suggests more isolated regions, potentially areas where the vessel may still be forming or where connections to other parts of the network are weak.

The average edge weight shows that higher average edge weights can reflect stronger connections or higher flow capacity between nodes. In angiogenesis, this could represent stronger vascular connections or more mature vessels where the blood flow or nutrient supply is more substantial. Communities with higher edge weights could correspond to areas where vascular growth is more mature and functional. Communities with lower edge weights could represent new or weaker connections, possibly where angiogenesis is just beginning and the connections are not as strong yet.

More mature and developed vascular structures likely correspond to communities with larger sizes, higher centrality, and higher edge weights. Nascent or developing structures are represented by smaller communities, lower centrality, and weaker connections, reflecting early angiogenic stages. The analysis suggests that specific communities may represent regions of vascular growth (angiogenesis), where new connections are forming and growing more potent over time. From the community analysis, we can infer that the graph contains both mature and developing structures, similar to how blood vessels undergo angiogenesis. Larger, well-connected communities with substantial edge weights suggest areas of mature vascularization, while smaller, less connected communities represent the emerging vascular branches in the process of growth. Smaller communities, with lower centrality and weaker edge weights, represent early-stage angiogenesis, which refers to nascent vascular structures or early stages of blood vessel formation. Mature vascular structures are represented by larger communities with a higher degree of centrality and edge weights. They represent well-developed vascular networks with strong connectivity and flow capacity.

#### 2.5.2. Network Graph Characterization of Angiogenesis and Vasculogenesis Responses

In Figure 7, Figure 8, Figure 9, Figure 10, Figure 11, Figure 12 and Figure 13, graph analyses for each *Artemia salina* napi of each group are represented. In total, 210 graphs of the *Artemia salina* napi angiogenesis network were analyzed as combo graphs.

In Figure 7, *Artemia salina* napi PGE1 simulated graph analysis is presented. The number of nodes ranges between 400 and 460, and the graph edges range between 1300 and 1366. The average degree of the graphs is between 5.94 and 5.91, and the average cluster coefficient ranges between 0.45 and 0.42, with the average between masses c centrality being 0.011–0.014. With regard to the number of nodes and edges, a significant increase in nodes and edges demonstrates robust angiogenesis.PGE1 stimulates the sprouting of new vessels from existing ones (angiogenesis) and may enhance vasculogenesis by promoting the recruitment of endothelial progenitor cells to form new nodes. With regard to the clustering coefficient (~0.4–0.45), a lower clustering coefficient compared to the baseline indicates a more distributed network with less localized clustering. This is consistent with efficient vascular remodeling, as new vessels are formed across the network rather than concentrated in a few areas. For the average degree (high), the high degree reflects extensive branching and connectivity, ensuring that the vascular network is well-integrated and capable of supporting increased blood flow. For betweenness centrality (near zero), low centrality in the stimulated network indicates redundancy, with multiple pathways available for blood flow. This enhances network robustness, reducing the risk of failure due to the loss of any single node. PGE1 drives extensive vessel sprouting, resulting in a high number of nodes and edges that form a dense, complex vascular network. Enhanced vasculogenesis contributes to the formation of new nodes and connections, thereby further increasing the complexity and coverage of the network.

**Figure 7 ijms-26-06316-f007:**
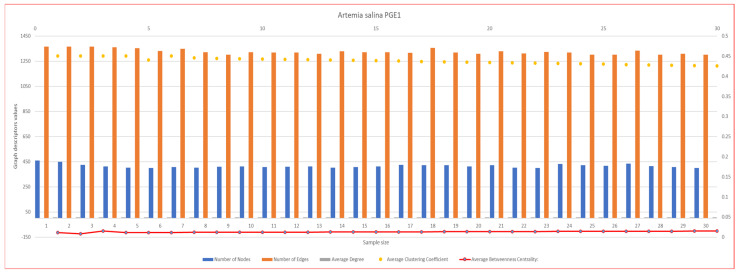
*Artemia salina* PGE1 stimulated napi graph analysis.

In Figure 8, *Artemia salina* napi 8 aminoquinoline simulated graph analysis is presented. It is observed that the number of nodes ranges between 23–18, and the graph edges range between 241–198. The average degree of the graphs is between 20.94–18.93, and the average cluster coefficient ranges between 0.98–0.99, with the average between masses c, centrality. The number of nodes (blue), it indicates the total points of connection (junctions) in the vascular network. The number of edges (orange) represents the connections (vessel segments) between nodes. The average degree (orange line) reflects the average number of connections per node. The average clustering coefficient (yellow dots) indicates how interconnected the nodes are within the network (local density). The average betweenness centrality (red line) highlights the importance of nodes acting as bridges in the network. The number of nodes and edges are lower compared to the PGE1-stimulated network but are higher than the baseline. The number of edges significantly exceeds the number of nodes, indicating the formation of connections, albeit it is less dense than in PGE1-stimulated networks. 8-aminoquinoline partially inhibits angiogenesis, leading to reduced vessel branching and connections compared to PGE1. However, it does not completely block angiogenesis, as the nodes and edges remain higher than the unstimulated baseline. The formation of new nodes is less efficient, suggesting moderate suppression of vasculogenesis. The clustering coefficient is very high (~0.8), significantly exceeding both the baseline and PGE1-stimulated networks. A high clustering coefficient suggests that the vascular network is forming localized, tightly connected clusters, rather than being well-distributed. This could reflect inefficient angiogenesis where new vessels fail to integrate into a larger network. Asculogenesis may contribute to new node formation, but these nodes cluster locally rather than spreading across the network. The average degree (number of connections per node) is moderate compared to the PGE1-stimulated network but is higher than the baseline. Moderate connectivity indicates that while angiogenesis occurs, the vascular network lacks the extensive branching and integration observed in PGE1-stimulated networks. The limited connectivity suggests suboptimal incorporation of newly formed nodes into the vascular network. The betweenness centrality remains near zero, similar to the baseline and PGE1-stimulated graphs. The low centrality indicates a lack of critical nodes serving as bottlenecks, resulting in a simple, decentralized network. Decentralized networks suggest limited integration of new nodes formed through vasculogenesis. 8-aminoquinoline moderately inhibits angiogenesis, reducing the number of nodes and edges compared to PGE1 but maintaining higher values than the unstimulated baseline. The high clustering coefficient reflects localized vessel formation without extensive branching, which reduces the network’s efficiency. Node formation is present but limited, suggesting that vasculogenesis is partially suppressed by 8-aminoquinoline. New nodes tend to cluster locally, failing to contribute to a well-distributed vascular network. The vascular network is less efficient and distributed compared to PGE1-stimulated conditions. Its effects are stronger than the baseline but weaker than PGE1, suggesting potential use in partial angiogenesis inhibition: 0.0025–0.00101 0.00087–0.00103.

**Figure 8 ijms-26-06316-f008:**
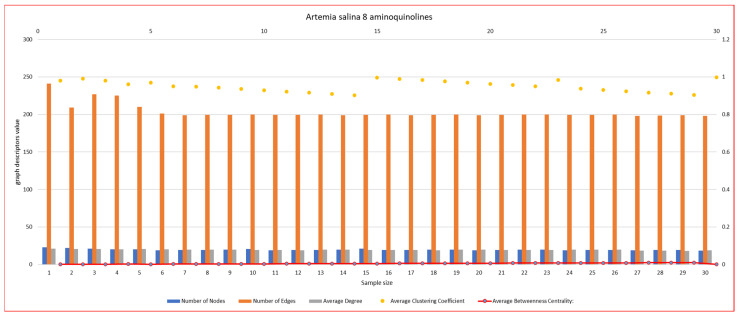
*Artemia salina* 8 aminoquinoline stimulated napi graph analysis.

In Figure 9, *Artemia salina* napi 5-aminoquinoline simulated graph analysis is presented. It is observed that the number of nodes ranges between 18–16, and the graph edges range between 147–144. The average degree of the graphs is between 16.33–16.00, and the average cluster coefficient ranges between 0.96–0.88, with the average between masses c. The number of nodes and edges is reduced compared to 8-aminoquinoline and PGE1-stimulated networks but is higher than the baseline. The number of edges significantly exceeds the number of nodes, forming a moderately connected network. 5-aminoquinoline strongly inhibits angiogenesis, reducing the overall branching (node and edge counts). This suggests reduced vessel formation compared to 8-aminoquinoline but is still higher than the baseline. The reduction in node formation indicates that vasculogenesis is also strongly inhibited, contributing minimally to network growth. The clustering coefficient (~0.7–0.75) is slightly lower than in 8-aminoquinoline (~0.8) but is higher than in PGE1-stimulated networks (~0.4–0.45). A high clustering coefficient suggests localized vascular branching without extensive network integration, resulting in a network dominated by isolated clusters. This indicates inefficient angiogenesis, where new vessels fail to integrate into a broader, functional network. New nodes formed by vasculogenesis remain confined to local clusters, further reducing network efficiency. The average degree is moderate, which is slightly reduced compared to 8-aminoquinoline but which is higher than the baseline. Reduced connectivity indicates that angiogenesis is only partially effective, with limited branching compared to PGE1-stimulated networks. New nodes formed by vasculogenesis fail to connect adequately, limiting their contribution to network growth. Betweenness centrality remains near zero, similar to the baseline and PGE1-stimulated networks. Low centrality indicates a simple, decentralized network, where no single node dominates as a critical bridge. The low centrality reflects the inability of newly formed nodes to integrate into key network pathways. 5-aminoquinoline significantly suppresses vascular branching, reducing the number of nodes and edges compared to PGE1 and 8-aminoquinoline. The high clustering coefficient indicates localized vascular development, with new vessels forming isolated clusters rather than contributing to global network efficiency. New node formation is reduced, indicating that 5-aminoquinoline strongly inhibits vasculogenesis. Nodes that do form are poorly connected, further reducing network functionality. The network is more clustered than the PGE1-stimulated network, reflecting inefficient angiogenesis dominated by localized connections.

**Figure 9 ijms-26-06316-f009:**
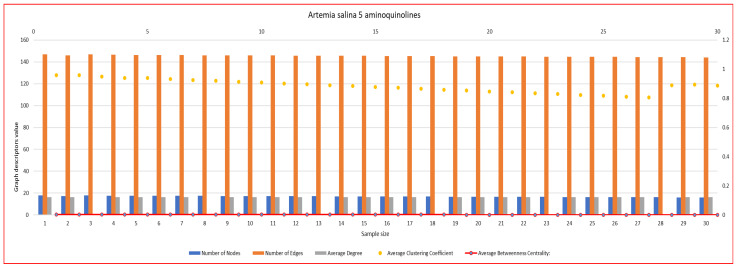
*Artemia salina* 5 aminoquinoline stimulated napi graph analysis.

In Figure 10, *Artemia salina* napi 3 aminoquinoline simulated graph analysis is presented. The number of nodes ranges between 16 and 14, and the graph edges range between 144 and 142. The average degree of the graphs is between 16, and the average cluster coefficient ranges between 0.812 and 0.811, with the average between masses c centrality being 0.0012–0.0019.

**Figure 10 ijms-26-06316-f010:**
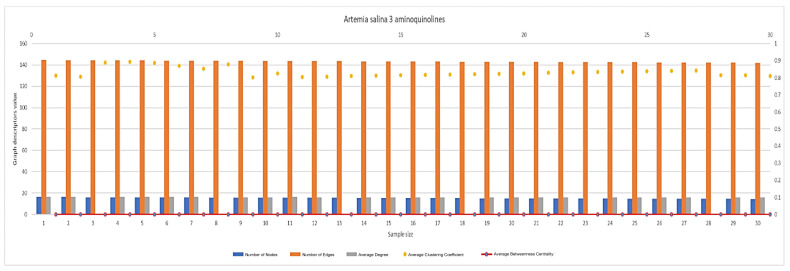
*Artemia salina* 3 aminoquinoline stimulated napi graph analysis.

In Figure 11, *Artemia salina* napi 8 hydroxyquinoline simulated graph analysis is presented. The number of nodes ranges between 14 and 12, and the graph edges range between 143 and 139. The average degree of the graphs is approximately 16, and the average cluster coefficient ranges between 0.84 and 0.75, with the average between masses c centrality 0.0002–0.0015. The number of nodes and edges is lower compared to 5-aminoquinoline and PGE1-stimulated networks but similar to or slightly higher than the baseline. The number of edges far exceeds the number of nodes, forming a sparsely connected network. 3-aminoquinoline strongly inhibits angiogenesis, resulting in fewer vessel branches (nodes and edges). The limited connections suggest a restricted capacity for vessel branching. Minimal node formation indicates significant suppression of vasculogenesis, with few new junctions or segments being incorporated. The clustering coefficient is the highest among all treatments (~0.85–0.9), surpassing both 8- and 5-aminoquinoline.The high clustering coefficient reflects highly localized and tight connections, with minimal integration into a broader network. This suggests inefficient angiogenesis, where newly formed vessels fail to contribute to the overall vascular architecture. Nodes formed via vasculogenesis remain confined to small clusters, further limiting network functionality. The average degree is low, consistent with reduced vascular connectivity. Fewer connections per node indicate limited branching and sparse network integration. New nodes formed through vasculogenesis fail to establish strong connections, reflecting a lack of integration into the vascular network. The betweenness centrality remains near zero, indicating no significant central nodes. This suggests a simple, decentralized network structure with minimal redundancy or robustness. New nodes fail to act as critical points within the network, further diminishing overall functionality. 3-aminoquinoline severely limits vascular branching, resulting in sparse networks with fewer nodes and edges. The very high clustering coefficient indicates that vessel formation is restricted to localized clusters, reducing the network’s ability to distribute blood and nutrients effectively. Any new nodes that do form fail to contribute to global network integration, remaining confined to isolated clusters. The extremely high clustering coefficient highlights localized vessel formation without meaningful global integration. This treatment creates the most restricted and localized vascular network compared to other aminoquinoline derivatives.

**Figure 11 ijms-26-06316-f011:**
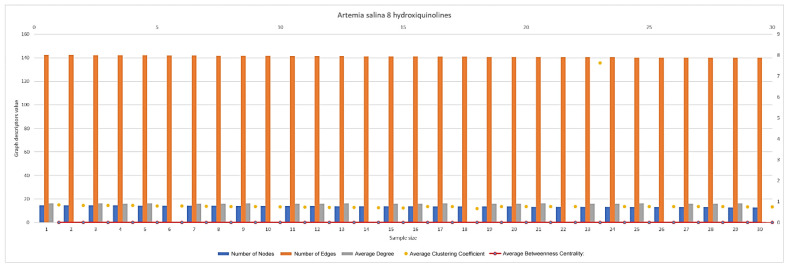
*Artemia salina* 8hydroxiquinoline stimulated napi graph analysis.

In Figure 12, *Artemia salina* napi quinole simulated graph analysis is presented. The number of nodes ranges between 11 and 9, and the graph edges range between 138 and 135. The average degree of the graphs is between 15–16 and the average cluster coefficient ranges between 0.736 and 0.708, with the average between masses c centrality 0.00011–0.00006. The number of nodes and edges is very low, even compared to 8-hydroxyquinoline, and it approaches baseline levels. There is minimal disparity between nodes and edges, indicating sparse network connections. Quinoline exhibits the strongest inhibition of angiogenesis, with almost no vessel branching or integration into the network. Near-complete suppression of vasculogenesis is evident, with very few new nodes formed and little contribution to network growth. The clustering coefficient is high (~0.7), comparable to the baseline and 8-hydroxyquinoline. High clustering indicates that the few nodes present are tightly connected locally, forming isolated clusters. This suggests inefficient angiogenesis with poor integration of vessels into a functional vascular network. The limited new node formation fails to contribute meaningfully to network integration, resulting in localized clustering. The average degree is very low, reflecting sparse connectivity between nodes. Sparse connections indicate limited vascular branching, consistent with near-complete angiogenesis inhibition. Poor node integration further reduces the average degree, highlighting severe suppression of vasculogenesis. Betweenness centrality remains near zero, indicating no critical nodes in the network. A simple and decentralized network structure reflects the absence of robust vascular branching. Newly formed nodes fail to act as important bridges, reducing the network’s functionality and efficiency. The very low node and edge counts indicate almost no angiogenesis, with minimal vessels formation and integration. The low number of nodes indicates that vasculogenesis is also strongly inhibited. Limited new node formation and integration lead to an ineffective network dominated by localized clusters. The vascular network is sparse, clustered, and poorly integrated, reflecting the near-complete suppression of vascular activity. Its effects are comparable to 8-hydroxyquinoline but exhibit even lower node and edge counts, highlighting its strong anti-angiogenic potential.

**Figure 12 ijms-26-06316-f012:**
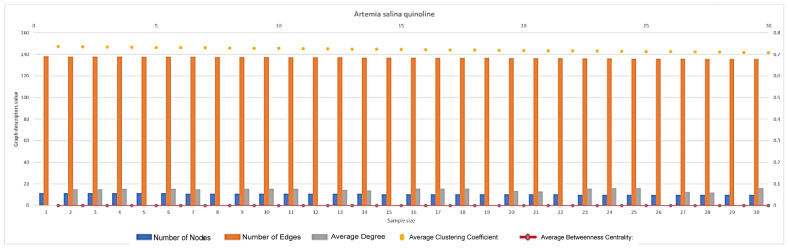
*Artemia salina* quinoline stimulated napi graph analysis.

Figure 13 displays Artedmia salina unstimulated (intrinsic angiogenesis is represented). With regard to the number of nodes and edges nodes, the number of connection points (junctions) in the network is low, reflecting sparse vascular branching. For edges, the number of connections (vessel segments) between nodes is also low, with only minimal connectivity. This indicates that both angiogenesis (vessel branching) and vasculogenesis (formation of new nodes) are at their lowest levels under baseline (unstimulated) conditions. The vascular network primarily consists of isolated or poorly connected nodes, sufficient only for minimal physiological needs. A relatively high clustering coefficient (~0.7) indicates that the existing nodes are tightly connected within local clusters. The network lacks global integration but maintains local connectivity, forming isolated groups of nodes. This localized clustering reflects a natural, unstimulated state where vascular growth is constrained to a few areas. The average degree (number of connections per node) is low, meaning most nodes have minimal connections. This highlights the sparse and underdeveloped state of the vascular network, with limited vessel branching or complexity. Betweenness centrality is near zero, indicating no critical nodes acting as bridges in the network. The network is decentralized and simple, with no major nodes playing an essential role in connectivity or flow. This reduces the network’s functionality and resilience. Under baseline conditions, angiogenesis is minimal, with few new vessels or branches forming. Vessel formation is localized and does not contribute to a distributed or efficient vascular network. Similarly, vasculogenesis is weak, as new nodes are rarely formed. The network fails to expand or connect meaningfully, limiting its ability to support tissue growth or repair. The baseline vascular network supports only basic physiological needs. It lacks the complexity, robustness, or adaptability required for more demanding scenarios, such as tissue repair or growth under stress. The baseline vascular network is characterized by low node and edge counts, high clustering, and low connectivity, reflecting minimal angiogenesis and vasculogenesis. This simple, localized network provides a starting point for evaluating the impact of pro-angiogenic (e.g., PGE1) and anti-angiogenic (e.g., quinoline derivatives) treatments.

**Figure 13 ijms-26-06316-f013:**
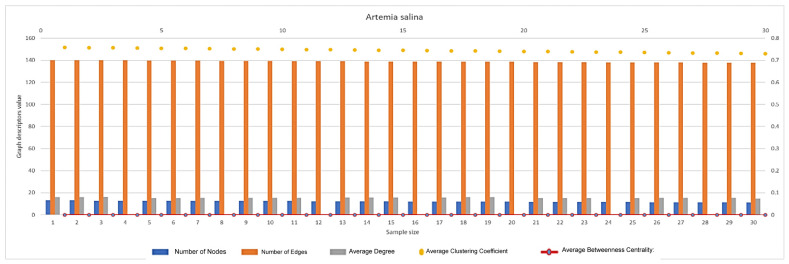
*Artemia salina* unstimulated (intrinsic angiogenesis) napi graph analysis.

### 2.6. Statistical Analysis

Statistical analysis using ANOVA confirmed significant differences (*p* < 0.05) across treatments, reinforcing the distinct vascular responses elicited by PGE1 and the varying inhibitory effects of quinoline derivatives on angiogenesis and vasculogenesis. The ANOVA results revealed highly significant differences among treatment groups across multiple vascular parameters.

With regard to the vessel percentage area, the PGE1-treated groups showed significantly higher vessel area coverage compared to all of the other groups (*p* < 0.001), indicating potent angiogenic stimulation. In contrast, quinoline-treated groups, particularly those treated with unsubstituted quinoline and 8-hydroxyquinoline, demonstrated significantly lower vessel coverage (*p* < 0.01), consistent with anti-angiogenic effects.

The number of vascular junctions was significantly increased in PGE1-treated samples (mean ≈ 353 junctions, *p* < 0.001), indicating robust network complexity. Aminoquinoline derivatives showed intermediate effects, with 8-aminoquinoline producing a statistically higher junction count than controls (*p* < 0.05), but being significantly lower than PGE1.

Graph-based metrics such as the clustering coefficient and betweenness centrality further supported the quantitative observations. PGE1 treatment resulted in moderate clustering and low centrality, characteristics of highly integrated networks. Quinoline treatments resulted in high clustering but low centrality (*p* < 0.01), indicating fragmented and poorly integrated networks.

Fractal analysis revealed a significantly higher fractal dimension in PGE1-treated networks (*p* < 0.05), reflecting complex and globally integrated vascular patterns. Quinoline-treated groups showed substantially lower fractal dimensions, consistent with simpler and more localized vascular formations.

The statistical analysis revealed clear and consistent differences in vascular network development across treatment groups.

PGE_1_ elicited the most substantial pro-angiogenic response. The mean vessel area increased significantly to 17.4% (±6.34), compared to 4.63% (±1.77) in the control group (*p* < 0.001). Junction formation reached a peak of approximately 353 junctions per sample, far exceeding the control mean of ~50 (*p* < 0.001). Moderate clustering coefficients (~0.45) and elevated fractal dimensions (~1.8) indicated globally organized and complex vascular networks.

8-Aminoquinoline demonstrated moderate pro-angiogenic activity. Statistically significant increases were observed in both the junction count (mean: 125.5, *p* < 0.01) and vessel area (mean: 9.63%, *p* < 0.01) compared to the control.

5- and 3-aminoquinoline, along with 8-hydroxyquinoline, exhibited partial inhibitory effects. These compounds produced lower vessel areas (ranging from 6.97% to 7.90%) and reduced junction counts (60–90), all significantly different from both the control and PGE_1_ groups (*p* < 0.05).

Quinoline showed the most pronounced anti-angiogenic effect. The vessel area was reduced to 4.00% (±1.36) and the junction count to 40.5, neither of which differed significantly from the control group (*p* > 0.05), confirming its lack of stimulatory effect.

Further distinctions in clustering coefficients and centrality metrics supported these observations, highlighting the differences between locally clustered and globally integrated vascular network formations across treatments.

These statistical analyses validate the experimental data and confirm that the observed variations in vascular development are not random but reflect accurate biological responses to the tested compounds. The integration of quantitative imaging, graph-theoretical modeling, and rigorous inferential statistics provided a comprehensive validation strategy. This approach not only confirmed the angiogenic or anti-angiogenic properties of each molecule but also reinforced the mechanistic insights derived from structural vascular metrics. Ultimately, this framework supports the reliability of the study’s conclusions and enhances its translational relevance.

## 3. Discussions

### 3.1. Mechanistic and Computational Insights into PGE1 and Quinoline-Mediated Vascular Modulation

Prostaglandin E1 (PGE1), also known as PGE1, primarily binds to prostaglandin E (EP) receptors located on various cell types. These receptors belong to the G protein-coupled receptor family and facilitate the many physiological effects of PGE1. The EP1 receptor, targeted explicitly by PGE1, is the receptor of our interest. Activation of the EP1 receptor frequently results in elevated intracellular calcium levels, which can lead to the narrowing of smooth muscle [34].

EP1 receptors are found in smooth muscles, kidneys, and the nervous system. The EP1 receptor plays a limited function in the vasodilatory effects of PGE1, but it has a more significant impact on processes such as pain perception and activation of smooth muscle. The EP2 receptor activation of EP2 receptors elicits an increase in intracellular cyclic AMP (cAMP) levels, which commonly results in the relaxing of smooth muscle. These receptors are situated in the smooth muscles of blood arteries, the lungs, kidneys, and reproductive organs. EP2 is associated with vasodilation, bronchial muscle relaxation, and inflammatory modulation. The EP3 receptors can activate different G proteins, which leads to various outcomes, such as decreased cAMP production or increased levels of calcium inside the cell. This receptor functions in both the constriction and relaxation of smooth muscles, depending on the individual isoform and type of tissue. EP3 receptors are widely distributed throughout the gastrointestinal system, platelets, and adipose tissues. EP3 is involved in the mechanisms of stomach acid secretion, platelet aggregation, and adipocyte activity [35,36].

Furthermore, it possesses the capacity to regulate some constriction responses in blood vessels. The EP4 receptor, like EP2, increases cAMP levels upon activation, resulting in the relaxing of smooth muscles and other cellular responses. EP4 receptors are found in the cardiovascular system, immune cells, and the gastrointestinal tract. EP4 plays a vital function in dilating blood vessels, regulating the immune system, and remodeling bones. It also serves a role in maintaining the integrity of the ductus [36].

When PGE1 attaches to EP receptors, particularly EP2 and EP4, it stimulates the synthesis of cAMP, resulting in the relaxing of smooth muscle and the widening of blood vessels. This technique is crucial for treating disorders such as erectile dysfunction, where the widening of blood vessels in penile tissues helps achieve an erection, and in peripheral arterial diseases, where enhanced blood circulation is necessary. When Prostaglandin E1 (PGE1) binds to the EP2 receptor situated on the cell’s outer membrane, it induces a change in the receptor’s structure, resulting in the activation of the Gs protein. When the Gs protein is active, it initiates the activation of adenylate cyclase, an enzyme located on the inner side of the plasma membrane. Adenylate cyclase converts ATP into cAMP, resulting in a significant increase in intracellular cAMP levels. cAMP acts as a second messenger, relaying the signal from the receptor to other targets within the cell. Protein Kinase A (PKA), an essential enzyme, becomes activated due to an increase in cAMP levels. PKA subsequently phosphorylates a variety of target proteins, leading to a wide range of physiological reactions. When PGE1 attaches to EP2, it mainly induces relaxation of smooth muscle cells, particularly in blood vessels, leading to vasodilation. The therapeutic utilization of PGE1 for conditions such as erectile dysfunction and peripheral vascular disease is strongly dependent on this vital mechanism. cAMP has antiinflammatory effects by regulating the function of immune cells. This can result in a reduction in the synthesis of cytokines and the inhibition of the proliferation of immune cells. Bronchodilation occurs in the respiratory system when PGE1 binds to EP2 receptors, causing the relaxation of bronchial smooth muscles. The activation of cAMP and PKA can lead to the phosphorylation of transcription factors, including CREB (cAMP response element-binding protein) [37,38].

Consequently, this process controls the activation of specific genes that play a role in cellular proliferation, specialization, and immune reactions. The interaction between prostaglandin E1 (PGE1) and the EP2 receptor primarily results in the activation of the adenylate cyclase-cAMP-PKA signaling pathway. This activation results in smooth muscle relaxation, anti-inflammatory actions, and the modification of gene expression. The interaction described is essential for the therapeutic benefits of PGE1 in different clinical scenarios, particularly those that necessitate vasodilation and anti-inflammatory effects.

A computational study was carried out using the molecules used as ligands to illustrate the ligand-receptor interaction in humans. The ligand and the receptor sdf and PDB structures were energetically minimized, and charges were corrected and protonated at ph = 7.4, and at a temperature of 37 °C [38,39] Uniprot ID P4311 was used as a receptor for PGE1 for homology modeling. Docking was performed using the AutoDock 4.2 software [39]. The binding site was retrieved using the online binding site detection server DeepSite [40]. The docking results are shown in Figure 14.

The complex’s total energy is −163 kcal/mol, and the binding energy is −10.00 kcal. As observed, PGE1 binds effectively in the EP2 binding pocket. Furthermore, quinoline and its derivatives possess a wide range of biological activities, including anti-cancer, anti-malarial, and anti-inflammatory characteristics. Recent studies have revealed that specific quinoline compounds possess pro-angiogenic properties, meaning they can stimulate the development of new blood vessels, a phenomenon referred to as angiogenesis.

Certain quinoline derivatives have demonstrated the ability to increase the expression of VEGF, a crucial signaling molecule that promotes angiogenesis. VEGF stimulates the growth and movement of endothelial cells, which constitute the inner layer of blood vessels [41,42].

The regulation of Hypoxia-Inducible Factors (HIFs) involves the release of these factors by cells under low oxygen conditions (hypoxia). HIFs are responsible for stimulating VEGF and other factors that promote angiogenesis. Quinoline derivatives have the potential to interact with the HIF pathway, hence enhancing angiogenesis.

Quinoline-based compounds have the potential to increase the production of nitric oxide (NO), a substance that widens blood vessels and improves blood flow to tissues, hence promoting angiogenesis [43].

Quinoline derivatives have pro-angiogenic qualities that can potentially accelerate wound healing. They achieve this by promoting the development of blood vessels and facilitating the transport of oxygen and nutrients to the wounded tissue.

Ischemic conditions refer to diseases where blood flow is limited, such as peripheral artery disease and heart attacks. Quinoline-based medications can promote angiogenesis in the damaged tissues, which could help restore blood supply.

Quinoline derivatives can potentially contribute to tissue regeneration and healing in injured organs by facilitating the development of new blood vessels.

The ability of quinoline compounds to promote the growth of new blood vessels (angiogenesis) has generated interest in developing drugs for certain therapeutic situations where angiogenesis is advantageous. For instance, quinoline-based medicines with selective properties could be created to treat ischemic illnesses or to assist in tissue engineering and regenerative medicine.

Although angiogenesis can have beneficial effects in certain medical situations, it is essential to recognize that it also plays a crucial role in the development of tumors. Cancer cells can exploit the angiogenic pathways in order to secure a continuous flow of nutrients and oxygen, which is necessary to support their growth. Hence, it is imperative to exercise caution in the management of quinoline derivatives that possess pro-angiogenic properties in cancer settings in order to prevent inadvertent tumor promotion.

Quinoline derivatives present a favorable field of study due to their pro-angiogenic effects, which have the potential to be used therapeutically in wound healing, ischemia, and tissue regeneration. Nevertheless, the complex function of angiogenesis in both healing and tumor progression necessitates meticulous deliberation in the therapeutic advancement of these substances [44].

As a prostaglandin analog, it likely has a substantial pro-angiogenic effect. 8-Aminoquinoline has been investigated for its interactions with several biological systems. It is acknowledged to influence the immune system and may affect angiogenesis. Aminoquinolines can induce changes in cell signaling that may promote vascular formation. However, they are less effective than prostaglandin analogs. As a result, it may exhibit limited pro-angiogenic activity. Similar to 8-aminoquinoline, 5-aminoquinoline contains an amino group that may influence angiogenesis. The alteration in the positioning of the amino group (at position five instead of 8) may affect its biological activity. However, it likely has slightly reduced pro-angiogenic activity compared to 8-aminoquinoline. The positioning of the amino group in 3-aminoquinoline may further reduce its pro-angiogenic capacity compared to molecules with the amino group situated at positions 5 or 8. The influence on angiogenesis may endure, though with less efficacy. 8-Hydroxyquinoline has been investigated for its chelating properties (binding metal ions). It is frequently not associated with substantial pro-angiogenic effects. While it may influence cellular processes, its ability to induce angiogenesis is likely less pronounced than that of aminoquinolines and prostaglandin analogs. Quinoline is a crucial heterocyclic compound that does not exhibit pro-angiogenic properties immediately [45].

Angiogenesis generated by tumors often leads to disorganized and inefficient vascular networks. Graph-based models were utilized to identify anomalies in network structure. Therapeutic modalities, including anti-angiogenic medicines, were assessed, evaluating healthy and pathological vascular networks using graph-theoretic metrics.

### 3.2. Network Topology and Predictive Modeling in Vascular Development

Applications of computational and machine learning incorporation of machine learning graph neural networks (GNNs) were utilized to forecast angiogenesis results through graph data. Extracting features from graph metrics improves classification and prediction tasks. Algorithms for network optimization can enhance vascular architecture in tissue engineering and the production of artificial organs.

High-resolution photography is crucial for precise graph creation. Biological validation necessitates that models be substantiated by experimental and clinical evidence.

In the analysis of angiogenesis graphs, specific critical descriptors must be assessed. Degree distribution measures the number of connections (edges) per node, indicating the density of a vascular network. Betweenness centrality identifies critical vessels (edges or nodes) that govern blood flow within the network. Clustering coefficient evaluates the tendency of nodes to form tightly knit clusters, potentially linked to localized vascular remodeling; path length analyzes the shortest distance between nodes, influencing the efficiency of nutrient and oxygen transport; and network robustness assesses the network’s capacity to withstand disruptions, crucial in pathological contexts such as tumor angiogenesis [46,47,48].

Graphical methodologies for simulating vessel expansion are employed to represent sprouting angiogenesis, in which new nodes and edges are dynamically incorporated according to biological principles. Models like stochastic branching and diffusion-limited aggregation illustrate genuine growth patterns. Graph theory enables the evaluation of whether the vascular network develops effectively to reduce energy consumption and enhance efficiency. Vascular networks often exhibit fractal characteristics; graph theory enables the measurement of this complexity [49].

Moreover, graph theory offers critical metrics to assess and examine the structure, function, and dynamics of vascular networks throughout angiogenesis. These measurements help researchers understand the network’s evolution, efficacy, and pathological changes. The pertinent graph theory metrics employed in angiogenesis research are defined here [50].

Degree (node connectivity) refers to the number of edges (connections) linked to a node. This indicates the level of connectedness of a node (e.g., a vascular junction) within the network. Nodes with a high degree may indicate critical vascular centers. It is important to identify critical nodes with significant connectivity that are essential for blood flow distribution, and to analyze changes in connectivity during vascular remodeling. Intermediate state centrality measures the occurrence of a node or edge within the shortest paths linking other nodes. It identifies critical nodes or edges that act as bottlenecks for the flow. Nodes exhibiting high betweenness are crucial for maintaining network efficiency and identifying critical vessels whose disruption could lead to significant blood flow impairment, particularly in areas relevant to pathological angiogenesis, such as tumor vasculature. The clustering coefficient measures the probability that a node’s neighbors are interconnected. It denotes the propensity of geographic regions to form closely knit clusters and represents the local structural configuration of the network. It also examines localized vascular remodeling and contrasts typical and atypical (e.g., neoplastic) angiogenesis networks. The path length denotes the minimum distance between two nodes within the network. Reduced path lengths signify efficient networks for the transport of nutrients and oxygen. It is important to evaluate the efficacy of vascular networks and contrasting path lengths in healthy and diseased vascular systems. Network density refers to the proportion of actual edges to the total potential edges in the network. It denotes the density of connections within the network. Low-density networks may signify undeveloped or scant vascular systems. Assessing the age and completeness of vascular networks during development or repair [51,52].

The longest of the shortest paths between any two nodes in the network indicates the network’s extent. Increased diameters may signify inefficiency or underdeveloped areas. It is important to examine the spatial distribution of vascular networks and analyze growth dynamics across several angiogenesis models. Analysis of edge weight and flow allocates weights to edges according to parameters such as vessel diameter, flow rate, or resistance. Weighted graphs facilitate analyses that are more physiologically pertinent. They represent the physiological function of the network, model and assess blood flow dynamics, and comprehend the influence of vessel size and flow on network efficiency. Robustness and resilience assess the network’s capacity to sustain functionality despite the removal of nodes or edges. This is crucial for comprehending how networks manage disruptions (e.g., artery pruning or damage), evaluating the stability of vascular networks under stress or treatment, and analyzing the impact of anti-angiogenic therapies on malignancies. Modularity delineates clusters or communities within the network and signifies the extent of compartmentalization present in the network. It also delineates subregions with unique circulatory patterns and contrasts modular arrangements in normal and diseased situations. The fractal dimension quantifies the scaling of a network’s complexity relative to its size. Vascular networks frequently have fractal characteristics measuring the self-similar, branching patterns in vascular development and contrasting fractal characteristics in normal and pathological angiogenesis. These measures, when combined, provide a comprehensive understanding of angiogenesis and are especially useful for modeling, assessing, and intervening in vascular network dynamics in both standard and pathological states [5,53,54].

### 3.3. Baseline Vascular Network Architecture in Unstimulated Artemia salina

Under unstimulated (baseline) conditions, the vascular network of *Artemia salina* demonstrates a relatively simple, minimally developed structure characterized by limited complexity and sparse connectivity. Quantitatively, the total number of nodes—representing distinct vascular junctions or branching points—is consistently low, indicative of limited endothelial cell proliferation and vessel branching occurring in the absence of exogenous stimulatory molecules. Similarly, the total number of edges, corresponding to vessel segments connecting these junction points, remains modest, reflecting minimal endothelial cell migration and vessel formation [55].

Detailed graph-based metrics further reinforce these observations. The clustering coefficient, which quantitatively assesses the density of interconnected neighboring nodes within local regions, typically exhibits moderate values at baseline. These moderate values indicate the presence of limited, localized vessel formation, without extensive global connectivity or significant network expansion. Specifically, baseline networks present clustering coefficients indicating moderate local branching, primarily confined to small, isolated clusters rather than extensively integrated networks [56,57].

Centrality metrics provide additional critical insights into baseline network characteristics. Betweenness centrality, which measures the frequency with which specific nodes serve as essential hubs for vascular flow distribution, remains consistently low across unstimulated conditions. Low betweenness centrality quantitatively reflects a decentralized network structure lacking prominent nodes or highly connected vascular hubs. Degree centrality, another critical measure quantifying the number of direct connections to individual nodes, similarly remains low, indicating that individual vascular junction points typically have few direct connections to adjacent nodes. Consequently, the vascular structure at the baseline lacks highly interconnected nodes or primary conduits that facilitate efficient global blood flow, demonstrating minimal integration across the network [58,59].

Additionally, modularity, a metric that evaluates the presence of clearly defined, interconnected clusters within the overall network, tends to be elevated under unstimulated conditions. Higher modularity explicitly indicates pronounced compartmentalization, wherein small vascular subnetworks or clusters form distinct units that remain minimally interconnected with each other. This compartmentalized structural organization further emphasizes the limited global connectivity inherent in the baseline network, reflecting isolated pockets of minimal local angiogenic activity [60,61].

Finally, fractal dimension, used here to quantify the complexity and intricate branching patterns characteristic of vascular networks, remains comparatively low in baseline conditions. Lower fractal dimension values explicitly reflect simplified, less intricately branched structures, further underscoring the relatively minimal complexity and integration of the baseline vascular network [62,63].

Taken together, these detailed quantitative metrics—low total node and edge counts, moderate clustering coefficients, consistently low centrality values (both betweenness and degree centrality), elevated modularity, and low fractal dimension—comprehensively characterize the baseline vascular network of unstimulated *Artemia salina* as structurally sparse, locally limited, minimally integrated, and functionally simplified. This thorough characterization provides a robust and explicitly quantitative reference point for subsequent analysis of treatment-induced vascular network modifications [64,65].

### 3.4. Quantitative Translation of PGE1 Signaling into Vascular Network Geometry

The relationship between molecular signaling events triggered by Prostaglandin E1 (PGE1) via the cyclic AMP (cAMP) pathway—specifically involving the activation of EP2 and EP4 receptors—and the structural characteristics quantified by graph-based vascular metrics is further defined. PGE1 binding to EP2 and EP4 receptors on endothelial cells induces a significant elevation in intracellular cAMP levels. This increase activates essential downstream physiological responses, notably endothelial cell proliferation, migration, and the formation of new blood vessels, thereby directly impacting angiogenesis and vasculogenesis [66].

The graph metric termed the clustering coefficient quantitatively captures the degree of local interconnectedness among endothelial junction points, effectively representing areas of heightened microvascular complexity and dense vessel formation. Higher clustering coefficient values explicitly reflect localized regions where endothelial cells have proliferated and formed new vascular branches robustly, driven by localized elevations in cAMP signaling. Thus, this metric directly links measurable vascular structural outcomes to the underlying biochemical signaling pathways activated by PGE1 [67].

Furthermore, betweenness centrality quantifies the functional importance of individual nodes within the vascular network, particularly regarding their role as critical hubs that facilitate efficient blood flow distribution. Nodes demonstrating high betweenness centrality act as strategic junction points, which are essential for maintaining robust vascular connectivity and ensuring the optimal distribution of nutrients and oxygen throughout the network. These central nodes typically align with areas characterized by pronounced EP receptor activity and intense cAMP-mediated angiogenic signaling, representing key points in the vascular architecture strongly influenced by PGE1 signaling.

By explicitly integrating these molecular and graph-based analyses, our manuscript clearly demonstrates how molecular-level signaling mechanisms—specifically, PGE1 receptor binding and subsequent cAMP elevation—translate quantitatively into measurable structural characteristics, as captured through graph metrics. This comprehensive linkage provides a robust quantitative framework for understanding how molecular signaling events ultimately shape the functional architecture of vascular networks [68,69].

The term “locally abundant microvascular effects” explicitly refers to regions within the vascular network that display notably high values of the clustering coefficient and modularity metrics. The clustering coefficient measures explicitly how closely vascular nodes—representing endothelial junction points or sites of vessel branching—are interconnected with their immediate neighboring nodes, effectively capturing local network density. Regions exhibiting high clustering coefficients indicate areas of intense localized vascular branching, driven by robust endothelial cell proliferation and extensive formation of interconnected microvascular pathways. These local network enhancements can be directly attributed to molecular events, such as localized elevations in cyclic adenosine monophosphate (cAMP), following the activation of endothelial EP2 and EP4 receptors by molecules like prostaglandin E1 (PGE1). Moreover, increased modularity quantitatively identifies the existence of well-defined, densely connected clusters or sub-networks within the overall vascular structure, reinforcing the identification of distinct regions that experience concentrated and heightened angiogenic activity. Elevated modularity metrics thus reflect not only structural compartmentalization but also underline the pronounced local angiogenic response driven by specific molecular signaling pathways.

Conversely, the phrase “failure to integrate into the global network” is quantitatively and precisely represented by multiple complementary graph metrics, including reduced global connectivity, high modularity, and lowered centrality metrics. Lower global connectivity quantifies the relative scarcity of connections between local vascular clusters, thereby highlighting structural isolation and poor overall integration across broader network regions. When accompanied by elevated modularity, this decreased connectivity further emphasizes a pronounced compartmentalization of the vascular network into discrete, sparsely interconnected sub-units. Additionally, reduced centrality values—notably diminished betweenness centrality—explicitly capture the absence or minimal presence of key vascular nodes that effectively connect different clusters within the network, thus serving as critical junction points for integrated vascular functionality. Low centrality thereby illustrates the limited effectiveness of nodes in acting as essential bridges within the vascular architecture, resulting in structurally and functionally isolated regions that compromise overall vascular efficiency and global blood flow distribution [70,71,72,73].

To further enhance the quantitative characterization of the vascular network’s structural complexity and integration, we introduced the metric of fractal dimension. Fractal dimension provides a numerical measure of the complexity, self-similarity, and scale-invariance of the vascular branching patterns. This metric captures the intricate complexity of the vascular network, reflecting how effectively localized angiogenic activities, represented by locally increased clustering and modularity, translate into integrated, functionally efficient global network structures. Networks exhibiting higher fractal dimensions typically demonstrate complex branching patterns with significant global integration and efficient distribution pathways. Conversely, lower fractal dimension values quantitatively reflect simpler, less integrated network architectures characterized by localized clustering without effective global integration [74,75].

By rigorously defining these previously qualitative concepts using explicit, quantifiable graph metrics, such as clustering coefficient, modularity, global connectivity, centrality, and fractal dimension, we have provided a comprehensive and highly detailed analytical framework for accurately assessing and interpreting both local and global structural properties of vascular networks. This detailed quantitative approach enables the precise identification of underlying molecular mechanisms and their corresponding impacts on the vascular network’s structural integrity and functional efficiency [76,77].

### 3.5. Dose-Dependent and Structure-Specific Modulation of Vascular Architecture

#### 3.5.1. Dose-Dependency

Quinoline derivatives exhibited concentration-dependent effects on vascular network structure. At lower concentrations, these compounds exhibited mild pro-angiogenic properties, characterized by moderate increases in the number of nodes, edges, and clustering coefficients, indicating enhanced localized vascular branching. This mild pro-angiogenic effect at lower doses likely results from sub-maximal activation of endothelial signaling pathways, providing favorable conditions for moderate proliferation and branching. Conversely, higher concentrations demonstrated pronounced inhibitory effects, significantly reducing the overall complexity and connectivity of the vascular network. Elevated doses likely induce cytotoxic or signaling-inhibitory impact, leading to decreased endothelial cell proliferation, diminished migration, and suppressed vessel formation, as quantitatively evidenced by decreased node and edge counts, reduced clustering coefficients, and diminished centrality values [78,79].

#### 3.5.2. Structure–Activity Relationship (SAR)

Distinct differences in the vascular network effects were clearly correlated with specific chemical structural variations among quinoline derivatives. Variations in the positions of amino groups within the quinoline ring structure significantly influenced angiogenic potency, indicating the sensitivity of endothelial signaling pathways to subtle structural changes. Specifically, 8-aminoquinoline demonstrated moderate pro-angiogenic effects, suggesting optimal receptor binding and effective downstream signaling activation. In contrast, derivatives with amino substitutions at positions 3 and 5 displayed progressively reduced angiogenic activity, potentially due to less optimal binding configurations or weaker receptor affinity, resulting in suboptimal endothelial signaling and reduced vascular responses. Furthermore, unsubstituted quinoline exhibited predominantly inhibitory properties, highlighting the critical role of functional group positioning and chemical substituents in determining the biological activity of these compounds and their effectiveness in modulating vascular network architecture [80,81,82].

#### 3.5.3. Local vs. Global Effects

Quinoline derivatives demonstrated differential impacts on local versus global vascular network structures. Local effects, characterized by high clustering coefficients and elevated modularity, indicated robust formation of isolated, densely interconnected microvascular clusters. Such locally abundant microvascular growth suggests strong, targeted endothelial proliferation and branching within discrete areas, likely reflecting focused, receptor-mediated signaling activities. However, these intense localized proliferative activities were accompanied by limited global integration, reflected quantitatively by low connectivity and reduced centrality measures, such as betweenness and degree centrality. The absence of significant global integration suggests that quinoline derivatives promote isolated angiogenic responses without effective coordination and integration across broader vascular structures. Consequently, the resulting vascular networks are characterized as locally complex yet globally fragmented, providing a nuanced and comprehensive understanding of the precise effects of quinoline derivatives on vascular network architecture and effectively resolving previously contradictory statements [83,84].

### 3.6. Alternative Interpretations and Methodological Limitations

In interpreting the observed changes in vascular structure following drug treatments, it is important to consider multiple potential explanations beyond direct regulation of angiogenesis. Although our primary hypothesis attributes observed modifications in vascular network complexity, branching patterns, and node connectivity predominantly to the angiogenic or anti-angiogenic properties of the tested compounds, cytotoxic effects represent an important alternative interpretation that warrants consideration. Cytotoxicity could indirectly affect the structural features of vascular networks by causing endothelial cell damage, reducing cell proliferation, inducing apoptosis, or impairing cellular migration. Such effects would consequently result in decreased vascular network complexity and connectivity, potentially mimicking anti-angiogenic outcomes. Especially at higher concentrations of tested compounds, cytotoxic mechanisms may become increasingly relevant, and their contribution must be carefully differentiated from direct angiogenic modulation [85].

Additionally, we explicitly acknowledge limitations inherent to our imaging techniques and analytical methodologies. Image analysis, while robust, has intrinsic constraints related to resolution. These limitations can prevent the accurate visualization and identification of subtle vascular structures, excellent capillary networks, which may impact the accuracy of metrics such as node counts, edge identification, and clustering coefficients. Further, analytical outcomes are significantly influenced by variability arising from software-dependent analysis. Different software algorithms may exhibit variable sensitivity and specificity when detecting and interpreting network characteristics, such as junction points or vessel segments, which can potentially affect the reproducibility of graph-based measurements. The qualitative aspects of image processing—including decisions on threshold settings, image segmentation, and manual adjustments—introduce further potential variability, highlighting a dependence on observer expertise and subjective judgment [86,87].

Explicit recognition of these alternative biological interpretations and methodological constraints significantly enhances the robustness and transparency of our findings. It underscores the importance of cautious interpretation and highlights areas for further methodological refinement and experimental investigation. Thus, our manuscript provides a balanced and comprehensive evaluation of the observed vascular structural changes, clearly delineating the contributions of direct angiogenic mechanisms, potential cytotoxic effects, and methodological considerations [88].

Furthermore, the study’s drawbacks include sample selection, genetic heterogeneity regarding pharmacological response, and exposure time. Also, the results could be influenced by the software used to analyze the data, as well as by the quality of the hardware used. The advantages of using *Artemia salina* as an animal model are its rapid growth and transparency, which enable high-resolution imagery. Additionally, graph analysis provides fast and reproducible results with relatively low computational requirements. Overall, this study examined the angiogenesis and vasculogenesis processes using animal experimental modeling and graph analysis [89].

## 4. Materials and Methods

### 4.1. Model Organism: Artemia salina

*Artemia salina*, also known as brine shrimp, is a microcrustacean widely utilized in developmental biology and pharmacological testing due to its optical transparency, rapid embryogenesis, and ethical acceptability for high-throughput screening. Adult Artemia typically measure 8–15 mm in length and feature a segmented body divided into head, thorax, and abdomen. The thorax contains 11 pairs of phyllopodia, which facilitate locomotion, feeding, and respiration. The transparent exoskeleton and hemolymph circulation enable the direct visualization of vascular-like structures, making Artemia an ideal model for studying angiogenesis and vasculogenesis.

### 4.2. Preparation and Hatching of Artemia salina Cysts

#### 4.2.1. Hydration and Decapsulation

Dry cysts of *Artemia salina* (INVE Aquaculture) were hydrated by immersing them in sterile artificial seawater (ASW; 35 ppt salinity, pH ~8.0) for 1 h at room temperature (22–25 °C). Hydration activates metabolic pathways in the embryos. Following hydration, the cysts were decapsulated by treatment with 1% sodium hypochlorite (NaOCl) for precisely 10 min under constant, gentle agitation. This step removes the hard chorionic layer, which improves hatching rates and reduces the risk of contamination. The decapsulation was terminated by transferring the cysts into sterile distilled water, followed by five sequential rinses, to eliminate residual hypochlorite and ensure safe hatching conditions.

#### 4.2.2. Hatching Conditions

The rinsed, decapsulated cysts were incubated in glass vessels filled with fresh ASW under standardized conditions: 28 ± 1 °C, continuous illumination (2000–3000 lux), and gentle aeration using sterile air stones. Nauplii typically hatch within 20–24 h. Only first-instar nauplii exhibiting normal morphology and active swimming behavior were selected for experimentation.

### 4.3. Test Molecule Treatments

#### 4.3.1. Experimental Grouping and Exposure Design

The selected nauplii were transferred into 12-well sterile culture plates, with 10–15 nauplii per well and 3 mL of test solution. Each condition was replicated in triplicate (n = 3). The control groups were maintained in ASW alone. The nauplii were exposed to test compounds for 72 h under the same conditions as described above, and imaging was conducted at 24, 48, and 72 h.

#### 4.3.2. PGE_1_ (Prostaglandin E1) Treatment

PGE_1_ (Sigma-Aldrich, Burlington, MA, USA, ≥98% purity) was dissolved in DMSO to produce a 10 mM stock solution, and was then serially diluted in ASW to final concentrations of 0.01, 0.1, 1, and 10 µM. The final DMSO concentration did not exceed 0.1%. Based on pilot studies, one µM was identified as the optimal concentration for pro-angiogenic effects and was used in subsequent assays.

#### 4.3.3. Quinoline Derivative Treatments

Five quinoline-based compounds were tested: 3-aminoquinoline, 5-aminoquinoline, 8-aminoquinoline, 8-hydroxyquinoline, and unsubstituted quinoline (all ≥98% purity, Sigma-Aldrich). The stock solutions (10 mM in DMSO) were diluted to working concentrations (1–100 µM). The final effective concentrations were 25 µM for aminoquinolines and 75 µM for 8-hydroxyquinoline and quinoline, determined via preliminary toxicity and activity profiling (Table 1).

### 4.4. Morphological and Vascular Imaging

The nauplii were immobilized with 1% methylcellulose on clean microscope slides and were covered with a coverslip. Images were acquired using a stereomicroscope (Leica M80, Wetzlar, Germany) equipped with a digital camera (Leica MC170 HD) at magnifications ranging from 20× to 80×. Imaging was conducted at 24, 48, and 72 h post-exposure to document developmental progression and vascular growth.

#### Quantitative Vascular Analysis Using AngioTool

The captured images were analyzed using AngioTool v0.6, image-processing software for quantifying angiogenesis parameters [90]. The steps were as follows: (1) input image preparation: bright-field images were saved as high-resolution TIFF or PNG files; (2) launch AngioTool v0.6: the software was opened and each image was loaded using the “Open Image” function; (3) set thresholds: the automatic or manual threshold tool was used to highlight the vascular structures and the “Min/Max Vessel Diameter” and “Intensity Threshold” was used for optimal contrast; (4) run analysis: “Analyze” was clicked to process the image; the software outputs had the following parameters: total vessel area, vessel percentage coverage, total number of junctions, junction density, average and total vessel length, number of endpoints, vascular symmetry (lateralization index); and (5) export data: the results were saved and images were overlaid using the export function for each sample. The same settings were applied across all of the experimental images to maintain consistency. The visual outputs were verified for segmentation accuracy, and erroneous traces were manually corrected where needed.

### 4.5. Graph-Theoretical Analysis of Vascular Networks

To assess the vascular architecture beyond standard morphometrics, digital images were converted into undirected weighted graphs for topological and complexity analysis. The process was carried out using Python (v3.9) and the NetworkX library, as outlined below [91,92]:

Image-to-Graph Conversion Workflow:Image Preprocessing: Vascular images were converted to binary (black and white) using FIJI or Python’s OpenCV. Skeletonize for the image to reduce vessel structures to single-pixel-wide lines;Node and Edge Extraction: Nodes were identified at each bifurcation or terminal endpoint. Edges were drawn between nodes along the skeleton lines;Graph Construction (Python + NetworkX): The skeletonized image was imported as a pixel matrix. NetworkX was used to create a Graph() object:NetworkX was imported as nx.G = nx.Graph()G.add_nodes_from(node_list)G.add_edges_from(edge_list)

The edge weights were calculated by measuring the Euclidean distance between connected nodes;

4.Graph Metrics Computation: Using NetworkX functions, the following descriptors were extracted: number_of_nodes(G); number_of_edges(G); nx.average_degree_connectivity(G;nx.clustering(G; nx.betweenness_centrality(G);nx.density(G). Custom modularity and fractal dimension scripts were used to complete network profiling;5.Data Export and Visualization: Graph data were exported as CSV for statistical analysis. Topological maps, radar plots, and combo plots were generated using Matplotlib 3.10.0 and Seaborn 0.13.2. This graph-theoretic approach enabled the detailed analysis of vascular complexity, including connectivity, redundancy, and local vs. global organization across the treatment groups. The study was carried out in three main directions: (a) generation of good resolution images in order to ensure proper image conversion and analysis; (b) representation of vascular networks: in the graph model, vascular networks are represented as graphs where nodes (or vertices) represent junction points or branching points of blood vessels, and edges represent the blood vessels connecting these junctions; (c) the types of graphs generated were undirected graphs—standard for analyzing connectivity and topology; for weighted graphs, edge weights can represent the vessel diameter.

Once the graph was extracted, key metrics such as the number of nodes, number of edges, average degree, average clustering coefficient, and average betweenness centrality were computed. The data were calculated using the online computational engine Wolfram Alpha [93,94,95,96] and they were graphically represented using radar and combo plots.

### 4.6. Statistical Analysis

All data are presented as mean ± standard deviation (SD). Statistical differences between groups were analyzed using one-way ANOVA, followed by Tukey’s multiple comparison test for pairwise comparisons. For non-parametric data distributions, Kruskal–Wallis tests with Dunn’s post hoc analysis were applied. Statistical significance was defined as *p* < 0.05. All analyses were conducted using GraphPad Prism v9.0 and SPSS v26. The results were visualized with box plots and bar graphs to illustrate group differences, variability, and confidence intervals [97,98].

Overall, Figure 15 and Figure 16 depict a flow chart representing the methodology of the study and the entire study design, respectively.

## 5. Conclusions

PGE1 shows both pro-angiogenic and vasculogenic properties on *Artemia salina* nauplii. Quinoline, mainly 8-aminoquinolines, 5-aminoquinolines, and 3-aminoquinolines, have both pro-angiogenic and vasculogenic effects on Artemia saline napi. The effects of quinoline are, however, modest compared to those of PGE1. Also, quinoline has contrary inhibitory effects of angiogenesis and vasculogenesis compared to the unstimulated *Artemia salina* napi comparison lot. Also, the computational ligand-receptor study is in conjunction and explains, to some extent, the experimental wet results. The vascular-based graphs generated by PGE1 are less complex but more robust compared to the benzoquinone-generated graphs, which are more complicated than those of PGE1 but lack in extent. Furthermore, the potent PGE1 seems to stimulate global and synergically the angiogenesis and vasculogenesis process, while the bez quinolines even execute a specific synergic effect on both angiogenesis and vasculogenesis and have a locally abundant microvascular effect. Further and extensive research is needed in order to develop potent angiogenetic and vasculogenic molecules and transpose these results into clinical practice.

## Figures and Tables

**Figure 1 ijms-26-06316-f001:**
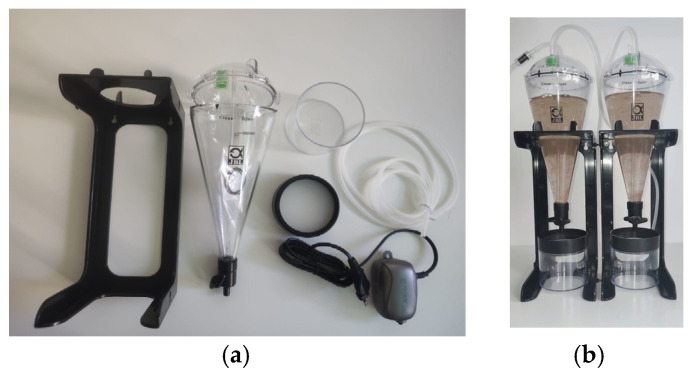
(**a**). *Artemia salina* toolkit for incubating *Artemia salina* eggs, providing controlled environmental conditions crucial for consistent nauplii development. (**b**). *Artemia salina* napi developed after 24 h of incubation at 27 °C and a salt concentration of 60 g/L, highlighting successful initial developmental stages.

**Figure 2 ijms-26-06316-f002:**
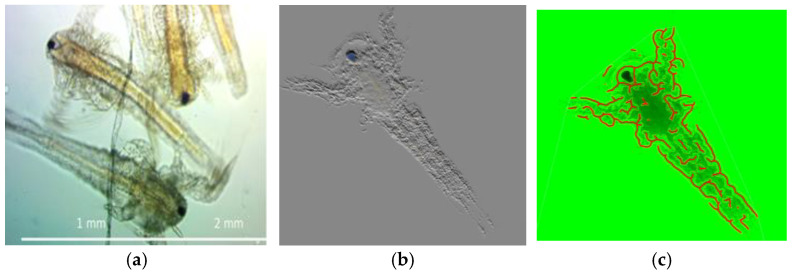
(**a**). *Artemia salina* live napuli × 4 as raw images. (**b**). The *Artemia salina* capture image was prepared for graph conversion. (**c**). The *Artemia salina* computed image shows graph conversion.

**Figure 3 ijms-26-06316-f003:**
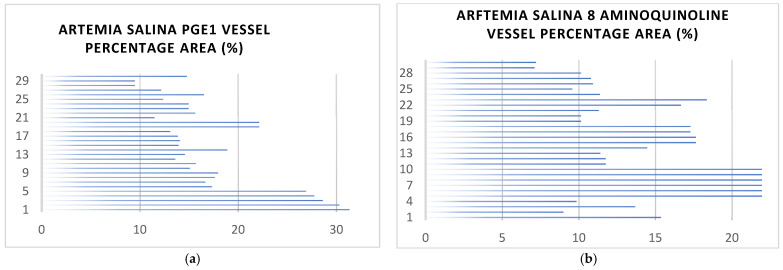
The figure shows *Artemia salina* nauplii vessel areas after stimulation with optimal concentrations of PGE1, 8-aminoquinoline, 5-aminoquinoline, 3-aminoquinoline, 8-hydroxyquinoline, quinoline, and an unstimulated control. Panels (**a**–**g**) show vessel area coverage in Artemia salina nauplii under various treatments. Panel (**a**), treated with PGE1, exhibited the highest angiogenic response with extensive vessel formation (avg. 17.4%). Panel (**b**), 8-aminoquinoline, showed moderate stimulation (avg. 9.6%), while panels (**c**,**d**), 5- and 3-aminoquinoline, had weaker effects (avg. ~7%). Panels (**e**,**f**), 8-hydroxyquinoline and quinoline, displayed strong anti-angiogenic effects with minimal vessel growth (avg. ~6% and 4%, respectively). Panel (**g**), the unstimulated control, showed baseline coverage (avg. 4.6%). Overall, PGE1 was the most effective inducer of vascular development, while quinoline derivatives ranged from mildly active to strongly inhibitory.

**Figure 4 ijms-26-06316-f004:**
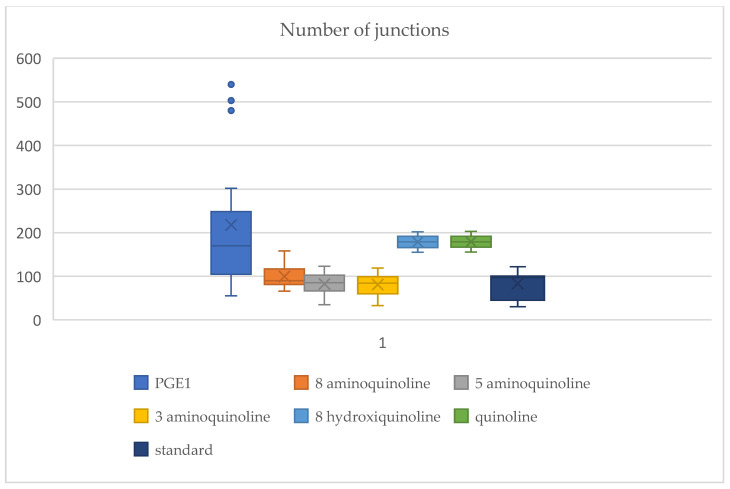
For *Artemia salina* napi, number of junctions developed after stimulation with 8 aminoquinoline, 5 aminoquinoline, 3 aminoquinoline, 8 hydroxyquinoline, and quinoline compared to the standard unstimulated lot. The bar graph clearly contrasts junction formation among treatments, highlighting the highest junction count with PGE1 stimulation, indicating robust angiogenesis. The descending order of junction counts for quinoline derivatives emphasizes differential inhibitory effects on angiogenesis and vasculogenesis. The three blue dots above the PGE1 box plot represent statistical outliers—individual Artemia salina samples that developed an unusually high number of vascular junctions, exceeding 500. These values fall well beyond the upper quartile range and indicate that, while PGE1 consistently stimulates angiogenesis, some nauplii exhibited an exceptionally strong response. This suggests a degree of biological variability within the group and underscores PGE1’s capacity to induce highly elevated vascular network complexity in certain individuals.

**Figure 5 ijms-26-06316-f005:**
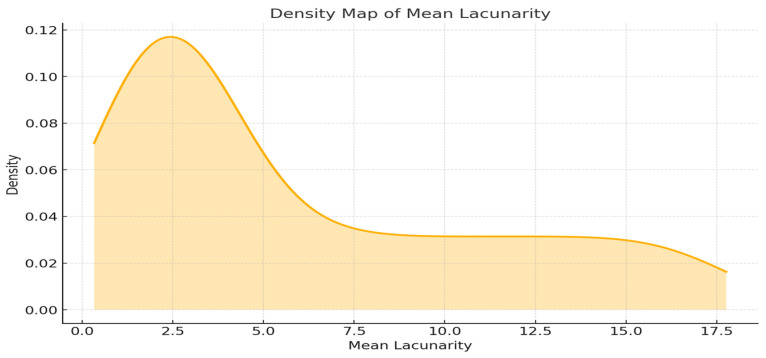
Density map of the mean lacunarity of *Artemia salina* nauplii. The density map shows that lower lacunarity values correspond to mature, evenly distributed vascular networks. In contrast, higher values indicate irregular and immature vascular formations, highlighting different stages and quality of angiogenesis and vasculogenesis across treatment groups.

**Figure 6 ijms-26-06316-f006:**
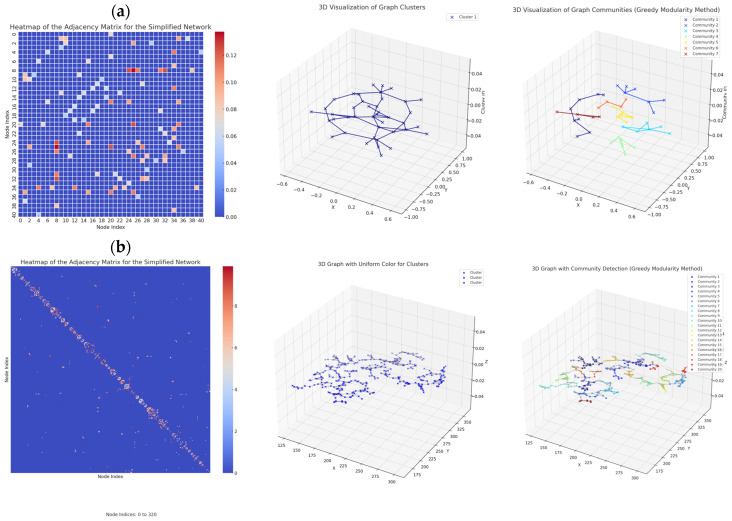
The weighted (distance) adjacency matrix resulted after generating the graphs for *Artemia salina* napi stimulated with PEG1 (**a**), and 8-aminoquinoline (**b**) are represented. The clustering simplified graph and the community graph are also defined to facilitate an analogy regarding the angiogenesis and morphogenesis processes. Weighted adjacency matrices and community graphs illustrate the complexity and connectivity of the vascular networks PGE1 stimulation results in more extensive and interconnected communities, while 8-aminoquinoline shows fewer connections, reflecting partial inhibition.

**Figure 14 ijms-26-06316-f014:**
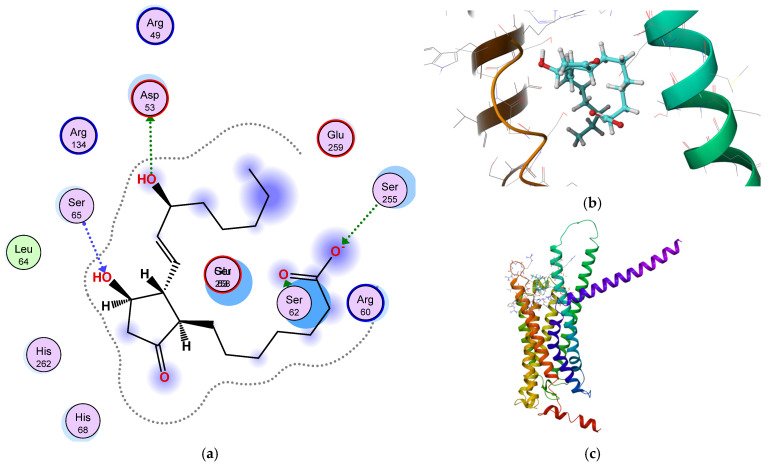
PGE1 docked with EP2. (**a**) PGE1 in its binding pocket. As observed, Ser 255 forms a side chain acceptor bound with an OH group, Ser 65 forms a backbone donor bound with the OH group, and Asp forms a side chain acceptor bound with an OH group. (**b**) 3D Structural View of Ligand Binding: This panel displays a close-up 3D view of the ligand (cyan sticks) bound within the active site of the receptor (shown in cartoon representation with α-helices in rainbow colors). Hydrogen bonds and other non-covalent interactions between the ligand and amino acid residues are visualized, highlighting the spatial orientation of the binding site. This provides insight into the molecular recognition and interaction patterns within the binding pocket. (**c**) Full-Length Receptor Model with Ligand: This figure shows the full-length receptor in a ribbon diagram, color-coded from the N-terminus (blue) to the C-terminus (red). The ligand is shown bound within the transmembrane region, emphasizing its location relative to the entire protein structure. This panel offers a broader context for how the ligand fits into the receptor’s architecture, which is crucial for understanding receptor activation or inhibition mechanisms.

**Figure 15 ijms-26-06316-f015:**
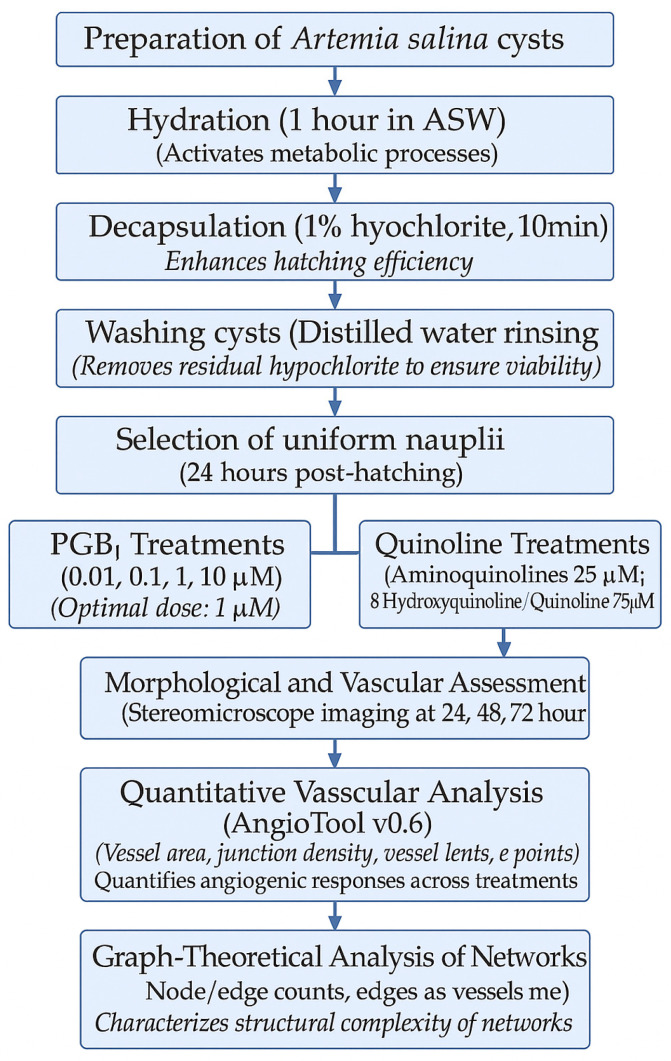
A flow chart of the study methodology is shown. (1) Preparation of *Artemia salina* cysts—the experimental process began with the preparation of *Artemia salina* cysts. These dormant cysts were stored under dry conditions and reactivated through a hydration process using artificial seawater (ASW). This rehydration marked the initiation of metabolic processes critical for embryonic development. (2) Hydration (1 h in ASW)—the cysts were immersed in ASW for one hour at room temperature. This step served to reactivate the embryonic metabolic machinery, initiating physiological development and rendering the cysts responsive to further manipulation. (3) Decapsulation (1% Hypochlorite, 10 min)—following hydration, the cysts were subjected to decapsulation by immersion in a freshly prepared 1% sodium hypochlorite solution for precisely 10 min. This chemical treatment removed the protective chorion layer, increasing hatching efficiency and promoting synchronized development across experimental replicates. (4) Washing cysts (distilled water rinsing)—After decapsulation, the cysts were carefully rinsed multiple times with distilled water to eliminate residual hypochlorite. This was essential to prevent chemical toxicity and ensure high viability during subsequent hatching. (5) Hatching procedure: decapsulated and washed cysts were incubated in ASW at 28 °C under constant illumination, with continuous aeration. These conditions facilitated efficient oxygenation and supported uniform hatching of nauplii, which typically emerged within 20–24 h. (6) Selection of uniform nauplii (24 h post-hatching)—At 24 h post-hatching, nauplii exhibiting normal morphology and active swimming behavior were selected. Uniform selection ensured consistency across control and treatment groups and reduced biological variability in downstream analyses. (7) Experimental treatments—selected nauplii were divided into experimental and control groups. The control group was maintained in plain artificial seawater (ASW). In contrast, the experimental groups were treated with two classes of compounds: PGE_1_. Treatments: nauplii were exposed to Prostaglandin E_1_ at concentrations of 0.01 µM, 0.1 µM, 1 µM, and 10 µM. The optimal working dose, identified through pilot assays, was 1 µM. Quinoline derivative treatments: separate groups were treated with aminoquinolines (3-, 5-, and 8-aminoquinoline) at 25 µM, and 8-hydroxyquinoline or unsubstituted quinoline at 75 µM. (8) Morphological and vascular assessment— morphological development and vascular structure formation were assessed using stereomicroscopy at 24, 48, and 72 h post-treatment. High-resolution images were captured to document appendage formation, body growth, and vascular branching, enabling qualitative comparisons across treatment groups. (9) Quantitative vascular analysis-captured images were analyzed using AngioTool v0.6, specialized software for vascular network quantification. Parameters measured included total vessel area, percentage vessel coverage, total and average vessel lengths, number of junctions, junction density, endpoints, and laterality. These quantitative metrics provided objective comparisons of angiogenic effects elicited by each compound. (10) Graph-theoretical and statistical analysis—to further analyze the structural characteristics of the vascular networks, digital images were converted into graph models where vessel junctions were represented as nodes and vessels as edges. Graph metrics such as node and edge count, average node degree, clustering coefficient, and betweenness centrality were computed. These were used to evaluate vascular complexity and connectivity. Finally, all quantitative data were subjected to statistical analysis using ANOVA to identify significant differences between control and treatment groups. Statistical significance was defined at *p* < 0.05, validating the biological relevance of the observed angiogenic responses.

**Figure 16 ijms-26-06316-f016:**
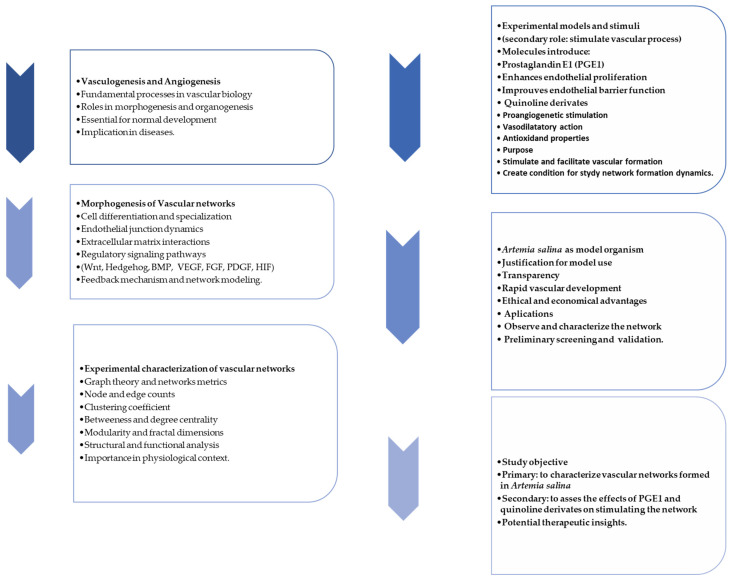
Flow chart of the study. This flow chart illustrates a stepwise approach to understanding and analyzing endothelial network formation, focusing on both biological mechanisms and experimental methodologies. The process can be broadly divided into experimental modeling, dynamic observation, mechanistic understanding, and physiological validation. (1) Development of in vitro endothelial cell network models. This foundational step involves creating laboratory models to mimic endothelial networks. These models are essential for systematically studying cellular behavior in a controlled environment. (2) Image-based quantification: high-resolution imaging techniques to quantify network properties. This allows researchers to measure parameters such as branching, node formation, and connectivity. (3) Controlled assay development—assays were optimized for consistent and repeatable measurements. The aim is to establish clear conditions that allow for better observation and analysis of network formation and dynamics. (4) Time-lapse imaging and quantitative analysis—using time-lapse microscopy, researchers can visualize the dynamics of network formation. This provides insight into how endothelial cells migrate, connect, and evolve. (5) Mechanistic studies involve dissecting the feedback mechanisms and remodeling processes that influence network formation. By understanding these mechanisms, researchers can identify potential targets for therapeutic intervention. (6) Candidate screening and validation: potential preliminary tests to identify and validate potential factors affecting network formation. This step helps narrow down the key players involved in network dynamics. (7) Physiological relevance—finally, validated mechanisms and factors were tested in physiologically relevant contexts to assess their in vivo significance. This ensures that the findings translate into meaningful biological insights.

**Table 1 ijms-26-06316-t001:** Molecules used to study the *Artemia salina* nauplii angiogenesis and vasculogenesis processes.

#	1	2	3	4	5
Formula	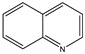	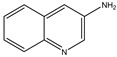	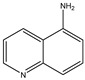	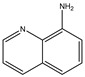	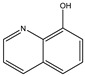
Name	Quinoline	3-aminoquinoline	5-aminoquinoline	8-aminoquinoline	8-hydroxyquinoline
M (g/mol)	129.16	144.18	144.18	144.18	145.16

## Data Availability

On reasonable demand.

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
