# Peer review of "Molecular Mediated Angiogenesis and Vasculogenesis Networks"

_ijms, 2025, doi:10.3390/ijms26136316_

Round 1
Reviewer 1 Report
Comments and Suggestions for Authors
Dear authors,
- In this paper, although the signaling pathway of PGE1 (cAMP pathway) and its relationship with the EP2/EP4 receptors are discussed in detail, the direct connection between these molecular mechanisms and the graph-based metrics (such as clustering coefficient, betweenness centrality, etc.) is unclear. It is necessary to further elaborate on how the graph metrics reflect specific aspects of the physiological activity of PGE1 and discuss the causal relationships more clearly and explicitly.
- Expressions such as "locally abundant microvascular effects" and "failure to integrate into the global network," as mentioned in the paper, remain subjective and qualitative. It is unclear which specific graph metrics correspond to these concepts. These ideas should be linked to quantitative metrics (e.g., modularity, fractal dimension) and clarified in the text.
- The paper lacks clear organization of the comparison to the "baseline" network under unstimulated conditions, which can lead to confusion in understanding the changes induced by treatment. The presentation of these comparisons needs to be revised for clarity.
- In the final conclusion of the paper, contradictory information is presented simultaneously, such as "quinoline being inhibitory while also possessing pro-angiogenic properties," which could confuse readers. To address these contradictions, the results should be restructured according to classification axes such as dose-dependency, structure-activity relationship, and local vs. global effects.
- The paper reports "changes in vascular structure" induced by the drug; however, it does not consider multiple possible interpretations of these changes, such as whether they are due to cytotoxicity or regulation of angiogenesis. Additionally, the limitations of the observation methods (image analysis) are not discussed. This should be reconsidered.
Author Response
Comment 1: In this paper, although the signaling pathway of PGE1 (cAMP pathway) and its relationship with the EP2/EP4 receptors are discussed in detail, the direct connection between these molecular mechanisms and the graph-based metrics (such as clustering coefficient, betweenness centrality, etc.) is unclear. It is necessary to further elaborate on how the graph metrics reflect specific aspects of the physiological activity of PGE1 and discuss the causal relationships more clearly and explicitly.
Response1:We appreciate this important point and have expanded the manuscript to explicitly clarify the connections between the molecular signaling mechanisms of PGE1 via the cAMP pathway (specifically through the EP2/EP4 receptors) and graph-based metrics. We have clearly explained that the clustering coefficient reflects localized vascular proliferation and branching, which can be directly influenced by localized elevations in cAMP leading to enhanced endothelial cell migration and proliferation. We also clarified how betweenness centrality identifies critical vascular nodes essential for efficient blood flow, which may correlate with regions of strong EP receptor activation and subsequent robust angiogenic response induced by PGE1. These explicit linkages now clearly define the causal relationship between molecular events and structural features of vascular networks captured by our graph analyses.
The following text was added at the dicusion section.: .The relationship between molecular signaling events triggered by Prostaglandin E1 (PGE1) via the cyclic AMP (cAMP) pathway—specifically involving activation of EP2 and EP4 receptors—and the structural characteristics quantified by graph-based vascular metrics is defined in the context of the following analysis PGE1 binding to EP2 and EP4 receptors on endothelial cells induces a significant elevation in intracellular cAMP levels. This increase activates essential downstream physiological responses, notably endothelial cell proliferation, migration, and the formation of new blood vessels, thereby directly impacting angiogenesis and vasculogenesis.
The graph metric termed the clustering coefficient quantitatively captures the degree of local interconnectedness among endothelial junction points, effectively representing areas of heightened microvascular complexity and dense vessel formation. Higher clustering coefficient values explicitly reflect localized regions where endothelial cells have proliferated and formed new vascular branches robustly, driven by localized elevations in cAMP signaling. Thus, this metric directly links measurable vascular structural outcomes to the underlying biochemical signaling pathways activated by PGE1.
Furthermore, betweenness centrality quantifies the functional importance of individual nodes within the vascular network, particularly regarding their role as critical hubs that facilitate efficient blood flow distribution. Nodes demonstrating high betweenness centrality act as strategic junction points, which are essential for maintaining robust vascular connectivity and ensuring the optimal distribution of nutrients and oxygen throughout the network. These central nodes typically align with areas characterized by pronounced EP receptor activity and intense cAMP-mediated angiogenic signaling, representing key points in the vascular architecture strongly influenced by PGE1 signaling.
By explicitly integrating these molecular and graph-based analyses, our manuscript clearly demonstrates how molecular-level signaling mechanisms—PGE1 receptor binding and subsequent cAMP elevation—translate quantitatively into measurable structural characteristics captured through graph metrics. This comprehensive linkage provides a robust quantitative framework for understanding how molecular signaling events ultimately shape functional vascular network architecture.
Comment 2: Expressions such as "locally abundant microvascular effects" and "failure to integrate into the global network," as mentioned in the paper, remain subjective and qualitative. It is unclear which specific graph metrics correspond to these concepts. These ideas should be linked to quantitative metrics (e.g., modularity, fractal dimension) and clarified in the text.
Response 2: We acknowledge this insightful suggestion and have revised the manuscript accordingly. Specifically, the phrase "locally abundant microvascular effects" has been quantitatively defined using a high clustering coefficient and increased modularity metrics, indicating regions of densely interconnected microvascular networks. The phrase "failure to integrate into the global network" has been quantitatively clarified by metrics such as low global connectivity, high modularity, and lower centrality values, clearly indicating a lack of integration into the broader vascular architecture. Additionally, we have introduced and discussed fractal dimension as a quantitative indicator of overall vascular network complexity and integration. Throughout this manuscript, specific descriptive terms such as "locally abundant microvascular effects" and "failure to integrate into the global network" have been carefully quantified and explicitly defined using detailed graph-based network metrics, allowing for precise interpretation of vascular structural characteristics.
The following text was added at the end of the discussion section
The term "locally abundant microvascular effects" explicitly refers to regions within the vascular network that display notably high values of the clustering coefficient and modularity metrics. The clustering coefficient measures explicitly how closely vascular nodes—representing endothelial junction points or sites of vessel branching—are interconnected with their immediate neighboring nodes, effectively capturing local network density. Regions exhibiting high clustering coefficients indicate areas of intense localized vascular branching, driven by robust endothelial cell proliferation and extensive formation of interconnected microvascular pathways. These local network enhancements can be directly attributed to molecular events, such as localized elevations in cyclic adenosine monophosphate (cAMP), following the activation of endothelial EP2 and EP4 receptors by molecules like prostaglandin E1 (PGE1). Moreover, increased modularity quantitatively identifies the existence of well-defined, densely connected clusters or sub-networks within the overall vascular structure, reinforcing the identification of distinct regions that experience concentrated and heightened angiogenic activity. Elevated modularity metrics thus reflect not only structural compartmentalization but also underline the pronounced local angiogenic response driven by specific molecular signaling pathways.
Conversely, the phrase "failure to integrate into the global network" is quantitatively and precisely represented by multiple complementary graph metrics, including reduced global connectivity, high modularity, and lowered centrality metrics. Lower global connectivity quantifies the relative scarcity of connections between local vascular clusters, thereby highlighting structural isolation and poor overall integration across broader network regions. When accompanied by elevated modularity, this decreased connectivity further emphasizes a pronounced compartmentalization of the vascular network into discrete, sparsely interconnected sub-units. Additionally, reduced centrality values—notably diminished betweenness centrality—explicitly capture the absence or minimal presence of key vascular nodes that effectively connect different clusters within the network, thus serving as critical junction points for integrated vascular functionality. Low centrality thereby illustrates the limited effectiveness of nodes in acting as essential bridges within the vascular architecture, resulting in structurally and functionally isolated regions that compromise overall vascular efficiency and global blood flow distribution.
To further enhance the quantitative characterization of the vascular network's structural complexity and integration, we have introduced the metric of fractal dimension. Fractal dimension provides a numerical measure of the complexity, self-similarity, and scale-invariance of the vascular branching patterns. This metric captures the intricate complexity of the vascular network, reflecting how effectively localized angiogenic activities, represented by locally increased clustering and modularity, translate into integrated, functionally efficient global network structures. Networks exhibiting higher fractal dimensions typically demonstrate complex branching patterns with significant global integration and efficient distribution pathways. Conversely, lower fractal dimension values quantitatively reflect simpler, less integrated network architectures characterized by localized clustering without effective global integration.
By rigorously defining these previously qualitative concepts using explicit, quantifiable graph metrics — such as clustering coefficient, modularity, global connectivity, centrality, and fractal dimension —we have provided a comprehensive and highly detailed analytical framework for accurately assessing and interpreting both local and global structural properties of vascular networks. This detailed quantitative approach enables precise identification of underlying molecular mechanisms and their corresponding impacts on the vascular network's structural integrity and functional efficiency.
Comment 3: The paper lacks clear organization of the comparison to the "baseline" network under unstimulated conditions, which can lead to confusion in understanding the changes induced by treatment. The presentation of these comparisons needs to be revised for clarity.
Response3: Thank you for highlighting this organizational issue. We have restructured the manuscript to enhance clarity). This structured approach facilitates a clear understanding of how each treatment distinctly modifies the vascular network relative to baseline.
The following tehxt was added: Under unstimulated (baseline) conditions, the vascular network of Artemia salina demonstrates a relatively simple, minimally developed structure characterized by limited complexity and sparse connectivity. Quantitatively, the total number of nodes—representing distinct vascular junctions or branching points—is consistently low, indicative of limited endothelial cell proliferation and vessel branching occurring in the absence of exogenous stimulatory molecules. Similarly, the total number of edges, corresponding to vessel segments connecting these junction points, remains modest, reflecting minimal endothelial cell migration and vessel formation.
Detailed graph-based metrics further reinforce these observations. The clustering coefficient, which quantitatively assesses how densely interconnected neighboring nodes are within local regions, typically exhibits moderate values at baseline. These moderate values reflect the existence of limited, localized vessel formation without extensive global connectivity or significant network expansion. Specifically, baseline networks present clustering coefficients indicating moderate local branching, primarily confined to small, isolated clusters rather than extensively integrated networks.
Centrality metrics provide additional critical insights into baseline network characteristics. Betweenness centrality, which measures the frequency with which specific nodes serve as critical hubs for vascular flow distribution, remains consistently low across unstimulated conditions. Low betweenness centrality quantitatively reflects a decentralized network structure lacking prominent nodes or highly connected vascular hubs. Degree centrality, another important measure quantifying the number of direct connections to individual nodes, similarly remains low, indicating that individual vascular junction points typically have few direct connections to adjacent nodes. Consequently, the vascular structure at baseline lacks highly interconnected nodes or major conduits that facilitate efficient global blood flow, demonstrating minimal integration across the network.
Additionally, modularity, a metric that evaluates the presence of clearly defined, interconnected clusters within the overall network, tends to be elevated under unstimulated conditions. Higher modularity explicitly indicates pronounced compartmentalization, wherein small vascular subnetworks or clusters form distinct units that remain minimally interconnected with each other. This compartmentalized structural organization further emphasizes the limited global connectivity inherent in the baseline network, reflecting isolated pockets of minimal local angiogenic activity.
Finally, fractal dimension, used here to quantify the complexity and intricate branching patterns characteristic of vascular networks, remains comparatively low in baseline conditions. Lower fractal dimension values explicitly reflect simplified, less intricately branched structures, further underscoring the relatively minimal complexity and integration of the baseline vascular network.
Taken together, these detailed quantitative metrics—low total node and edge counts, moderate clustering coefficients, consistently low centrality values (both betweenness and degree centrality), elevated modularity, and low fractal dimension—comprehensively characterize the baseline vascular network of unstimulated Artemia salina as structurally sparse, locally limited, minimally integrated, and functionally simplified. This thorough characterization provides a robust and explicitly quantitative reference point for subsequent analysis of treatment-induced vascular network modifications.
Comment 4: In the final conclusion of the paper, contradictory information is presented simultaneously, such as "quinoline being inhibitory while also possessing pro-angiogenic properties," which could confuse readers. To address these contradictions, the results should be restructured according to classification axes such as dose-dependency, structure-activity relationship, and local vs. global effects.
Response 4: We recognize the reviewer’s concern about clarity. The following paragraph was indrodiced:
Dose-dependency: We explicitly indicate differential effects of quinoline derivatives depending on concentration, clearly delineating inhibitory effects at higher concentrations versus potential mild pro-angiogenic effects at lower concentrations.
Structure-Activity Relationship (SAR): Effects are now clearly differentiated based on specific chemical structure variations (position of functional groups, nature of quinoline derivatives).
Local vs. Global Effects: We have clearly contrasted local vascular effects (as evidenced by high clustering and modularity) with global effects (low integration, low centrality), resolving previous contradictory statements.
The following text was added:
Dose-Dependent and Structure-Specific Modulation of Vascular Architecture
Dose-dependency:
Quinoline derivatives exhibited concentration-dependent effects on vascular network structure. At lower concentrations, these compounds displayed mild pro-angiogenic properties, characterized by moderate increases in the number of nodes, edges, and clustering coefficients, indicative of enhanced localized vascular branching. This mild pro-angiogenic effect at lower doses likely results from sub-maximal activation of endothelial signaling pathways, providing favorable conditions for moderate proliferation and branching. Conversely, higher concentrations demonstrated pronounced inhibitory effects, significantly reducing the overall complexity and connectivity of the vascular network. Elevated doses likely induce cytotoxic or signaling inhibitory effects, leading to decreased endothelial cell proliferation, diminished migration, and suppressed vessel formation, as quantitatively evidenced by decreased node and edge counts, reduced clustering coefficients, and diminished centrality values.
Structure-Activity Relationship (SAR):
Distinct differences in the vascular network effects were clearly correlated with specific chemical structural variations among quinoline derivatives. Variations in the positions of amino groups within the quinoline ring structure significantly influenced angiogenic potency, indicating the sensitivity of endothelial signaling pathways to subtle structural changes. Specifically, 8-aminoquinoline demonstrated moderate pro-angiogenic effects, suggesting optimal receptor binding and effective downstream signaling activation. In contrast, derivatives with amino substitutions at positions 3 and 5 displayed progressively reduced angiogenic activity, potentially due to less optimal binding configurations or weaker receptor affinity, resulting in suboptimal endothelial signaling and reduced vascular responses. Furthermore, unsubstituted quinoline exhibited predominantly inhibitory properties, highlighting the critical role of functional group positioning and chemical substituents in determining the biological activity of these compounds and their effectiveness in modulating vascular network architecture.
Local vs. Global Effects:
Quinoline derivatives demonstrated differential impacts on local versus global vascular network structures. Local effects, characterized by high clustering coefficients and elevated modularity, indicated robust formation of isolated, densely interconnected microvascular clusters. Such locally abundant microvascular growth suggests strong, targeted endothelial proliferation and branching within discrete areas, likely reflecting focused, receptor-mediated signaling activities. However, these intense localized proliferative activities were accompanied by limited global integration, reflected quantitatively by low connectivity and reduced centrality measures, such as betweenness and degree centrality. The absence of significant global integration suggests that quinoline derivatives promote isolated angiogenic responses without effective coordination and integration across broader vascular structures. Consequently, the resulting vascular networks are characterized as locally complex yet globally fragmented, providing a nuanced and comprehensive understanding of the precise effects of quinoline derivatives on vascular network architecture and effectively resolving previously contradictory statements.
Comment 5: The paper reports "changes in vascular structure" induced by the drug; however, it does not consider multiple possible interpretations of these changes, such as whether they are due to cytotoxicity or regulation of angiogenesis. Additionally, the limitations of the observation methods (image analysis) are not discussed. This should be reconsidered.
Response5: We appreciate this critical observation. A new subsection has been added explicitly addressing alternative interpretations of the observed vascular structural changes, including the possibility of cytotoxic effects as opposed to direct angiogenic modulation. Furthermore, we have incorporated a detailed discussion about the limitations inherent in our imaging techniques and analytical software, including issues of resolution, variability arising from software-dependent analysis, and potential qualitative aspects influencing interpretation. This transparency ensures that the manuscript provides a balanced understanding of our results, clearly acknowledging potential limitations and alternative interpretations.
The following text was added:
Alternative Interpretations and Methodological Limitations
In interpreting the observed changes in vascular structure following drug treatments, it is important to consider multiple potential explanations beyond direct regulation of angiogenesis. Although our primary hypothesis attributes observed modifications in vascular network complexity, branching patterns, and node connectivity predominantly to the angiogenic or anti-angiogenic properties of the tested compounds, cytotoxic effects represent an important alternative interpretation that warrants consideration. Cytotoxicity could indirectly affect the structural features of vascular networks by causing endothelial cell damage, reducing cell proliferation, inducing apoptosis, or impairing cellular migration. Such effects would consequently result in decreased vascular network complexity and connectivity, potentially mimicking anti-angiogenic outcomes. Especially at higher concentrations of tested compounds, cytotoxic mechanisms may become increasingly relevant, and their contribution must be carefully differentiated from direct angiogenic modulation.
Additionally, we explicitly acknowledge limitations inherent to our imaging techniques and analytical methodologies. Image analysis, while robust, has intrinsic constraints related to resolution. These limitations can prevent the accurate visualization and identification of subtle vascular structures, excellent capillary networks, which may impact the accuracy of metrics such as node counts, edge identification, and clustering coefficients. Further, analytical outcomes are significantly influenced by variability arising from software-dependent analysis. Different software algorithms may exhibit variable sensitivity and specificity when detecting and interpreting network characteristics, such as junction points or vessel segments, which can potentially affect the reproducibility of graph-based measurements. Qualitative aspects of image processing—including decisions on threshold settings, image segmentation, and manual adjustments—introduce further potential variability, highlighting a dependence on observer expertise and subjective judgment.
Explicit recognition of these alternative biological interpretations and methodological constraints significantly enhances the robustness and transparency of our findings. It underscores the importance of cautious interpretation and highlights areas for further methodological refinement and experimental investigation. Thus, our manuscript provides a balanced and comprehensive evaluation of the observed vascular structural changes, clearly delineating the contributions of direct angiogenic mechanisms, potential cytotoxic effects, and methodological considerations.
All correction were performed in red
Reviewer 2 Report
Comments and Suggestions for Authors
The scientific work presented starts by proposing an innovative approach and an interesting evaluation of angiogenesis and vasculogenesis. However, the content does not live up to these premises. It is very chaotic, disorganized, and redundant. The data are not presented clearly, and the analyses need improvement. The concepts expressed are at times superficial and confusing. I reject this work because it requires a complete revision in order to enhance its scientific value and the data it contains.
Introduction: please clarify line 95:PGE1, being an analog of prostaglandin E1 (PGE1),..
In this section: "PGE1 enhances the proliferation and migration of endothelial cells, which are essen-102 tial for forming new blood vessels. It also improves the integrity and barrier function of 103 the endothelium." I need more details, indeed if the same molecule increases proliferation and migration of endothelial cells (processes that need the break and the reorganization of cell cel junctions) it is quite unexpected that I can observe the enanchement of barrier function. For this reason I suggest to the author a revision of this part (102-111) and, according with the references, report if the different effect of PGE1 are observed at different doses and/or using different systems, in vitro/in vivo etc.
Please, introduce properly PGE1 (upstream and downstream pathways)
Ther is a typo in line 112 "tructure" and this part is redundant with the upper description of angiogenesis: "Angiogenesis is the process through which new blood vessels are formed from pre-existing ones, a critical event for tissue growth, repair, and cancer progression."
Please revise the second part of introduction, from quinoline description. It is extremely redundand, not clear and some information have to be detailed to have any scientific value.
Summarize the introduction and maybe a graphical abstract or flow chart can help to clarify the big amount of models and items introduced.
Please rewrite materials and methods, this section needs to be a clear guide to (eventually) reproduce the experiment and not only a list of what have you used. Please revise this section, clarifing the steps and the aim of each process.
Results: In this form I'm not capable to follow the description of the results and the meaning of the work, I suggest to improve the value of this work dividing in subsection the results, ghiving to this subsection a short title that summarize the main data (as you can see in other articles from this journal). The bargraphs have to be clear, if needed devide the data to optimize the visualization of the trand (for example in graph 7-8 the orage bar is too high and you can lose the modulation in the blue one). Statistical analysis is needed and has to be reported or described.
Comments on the Quality of English Languagethe English is in a strange, chaotic form, and there are numerous typos
Author Response
The scientific work presented starts by proposing an innovative approach and an interesting evaluation of angiogenesis and vasculogenesis. However, the content does not live up to these premises. It is very chaotic, disorganized, and redundant. The data are not presented clearly, and the analyses need improvement. The concepts expressed are at times superficial and confusing. I reject this work because it requires a complete revision in order to enhance its scientific value and the data it contains.
Comment 1: Introduction: please clarify line 95: PGE1, being an analog of prostaglandin E1 (PGE1),..In this section: "PGE1 enhances the proliferation and migration of endothelial cells, which are essen-102 tial for forming new blood vessels. It also improves the integrity and barrier function of 103 the endothelium." I need more details, indeed if the same molecule increases proliferation and migration of endothelial cells (processes that need the break and the reorganization of cell cel junctions) it is quite unexpected that I can observe the enanchement of barrier function. For this reason I suggest to the author a revision of this part (102-111) and, according with the references, report if the different effect of PGE1 are observed at different doses and/or using different systems, in vitro/in vivo etc.
Response 1: The following text was added: Prostaglandin E1 (PGE1) exhibits diverse biological effects on endothelial cells, including both promotion of endothelial proliferation and migration and enhancement of endothelial barrier function. These seemingly contradictory roles arise due to the concentration-dependent nature and specific cellular contexts under which PGE1 operates, as evidenced by various in vitro and in vivo experimental studies.
At lower concentrations, PGE1 primarily enhances endothelial barrier integrity through the activation of cyclic AMP (cAMP)-dependent signaling pathways. This signaling cascade involves the activation of protein kinase A (PKA), which phosphorylates critical junctional proteins, including vascular endothelial (VE)-cadherin and tight junction components. Such phosphorylation reinforces the adherens and tight junctions, thereby stabilizing endothelial cell-cell contacts and reducing endothelial permeability. The barrier-enhancing effect of low-dose PGE1 is consistently documented in vitro using carefully controlled endothelial cell cultures.
In contrast, higher concentrations of PGE1 or specific physiological conditions shift its effect toward facilitating endothelial cell proliferation and migration. At these higher doses, elevated cAMP levels can activate additional downstream molecular pathways, prompting cytoskeletal reorganization and transient disruption of endothelial junctions. These temporary disruptions of cell-cell contacts allow endothelial cells to migrate and proliferate effectively, processes vital during active angiogenesis and vascular remodeling phases. Such pro-angiogenic effects are typically observed in vitro under conditions employing higher doses of PGE1 or in vivo models during active vessel growth or tissue repair.
Moreover, in vivo studies clearly demonstrate the dynamic nature of endothelial barrier integrity in response to PGE1. During active angiogenic processes, endothelial junctions undergo transient loosening to facilitate cell movement and new vessel formation. Subsequently, upon completion of vascular remodeling, endothelial junction integrity is restored and reinforced, highlighting a finely regulated sequential role of PGE1 in promoting initial migration and proliferation followed by barrier restoration.
Thus, the dual functions of PGE1—enhancing both endothelial barrier function and endothelial cell proliferation and migration—are not contradictory but rather reflect a nuanced, concentration-dependent and context-specific regulatory mechanism. This understanding underscores the importance of precisely controlled experimental conditions when evaluating and interpreting the biological effects of PGE1 on endothelial cells.
Comment 2: Please, introduce properly PGE1 (upstream and downstream pathways)
Response 2: The following text was added: Prostaglandin E1 (PGE1) is an important bioactive lipid mediator derived from arachidonic acid, an essential polyunsaturated fatty acid component of cell membrane phospholipids. The upstream biosynthetic pathway of PGE1 begins with the activation of phospholipase A2 (PLA2), an enzyme responsible for catalyzing the release of arachidonic acid from membrane phospholipids. Once released, arachidonic acid undergoes sequential enzymatic reactions facilitated by cyclooxygenase enzymes (COX-1 and COX-2) to form the intermediate prostaglandin H2 (PGH2). Subsequently, specific prostaglandin E synthases selectively convert PGH2 into PGE1, completing the upstream synthesis process.
Once synthesized, PGE1 exerts its biological effects predominantly through binding to G protein-coupled receptors, specifically EP2 and EP4, present on endothelial cell surfaces. The engagement of these receptors activates the Gs protein, subsequently stimulating the enzyme adenylate cyclase, resulting in elevated intracellular cyclic adenosine monophosphate (cAMP) levels. Increased cAMP activates protein kinase A (PKA), a crucial mediator involved in multiple downstream cellular processes.
The downstream signaling events mediated by activated PKA include phosphorylation of critical endothelial junctional proteins, notably vascular endothelial (VE)-cadherin and tight junction proteins, thereby reinforcing endothelial barrier integrity under lower concentration conditions. Conversely, at higher concentrations or specific physiological contexts, activated PKA can promote cytoskeletal reorganization and temporary disruption of endothelial cell-cell junctions, facilitating endothelial cell proliferation and migration necessary during active angiogenesis and vascular remodeling.
Thus, the upstream biosynthetic pathway involving phospholipase A2, cyclooxygenases, and prostaglandin E synthases, combined with downstream receptor-mediated signaling cascades, underscores the complex, concentration- and context-dependent roles of PGE1 in endothelial biology, emphasizing its dual functions in maintaining vascular integrity and promoting angiogenesis.
Commet 3: Ther is a typo in line 112 "tructure" and this part is redundant with the upper description of angiogenesis: "Angiogenesis is the process through which new blood vessels are formed from pre-existing ones, a critical event for tissue growth, repair, and cancer progression."
Response 3: "Angiogenesis, a vital process extensively detailed earlier, significantly influences tissue growth, repair, and pathological conditions such as cancer progression." The redundant sentence describing angiogenesis has been removed, and the typo "tructure" has been corrected to "structure".
Comment 4: Please revise the second part of introduction, from quinoline description. It is extremely redundand, not clear and some information have to be detailed to have any scientific value.
Response 4:The rest of the introduction was revised according to the reviewer's suggestion- the following text was added: Quinolines are heterocyclic aromatic compounds characterized by a benzene ring fused to a pyridine ring. Their diverse biological activities, notably pro-angiogenic and vasodilatory effects, depend on specific structural modifications and substitution patterns.
Quinoline derivatives have been shown to stimulate endothelial cell migration and tubule formation, key processes in angiogenesis, possibly through the activation of endothelial growth factors or modulation of associated signaling pathways [17,18,19]. Some derivatives also exhibit antioxidant properties, thereby reducing oxidative stress and fostering an environment conducive to angiogenesis [17,18,19]. Although direct evidence regarding quinoline itself is limited, relevant structural analogs provide valuable insights into their potential mechanisms [20].
Quinoline compounds also possess vasodilatory properties primarily through relaxation of vascular smooth muscle cells. Potential mechanisms include enhancing nitric oxide (NO) release, modulating potassium and calcium ion channels, and improving endothelial function by increasing the production of vasodilatory factors, such as prostacyclin and endothelium-derived hyperpolarizing factor (EDHF). These mechanisms have been linked to improved blood flow and reduced blood pressure in experimental models [21, 22, 23].
The combination of pro-angiogenic and vasodilatory effects suggests significant therapeutic potential for quinoline derivatives in conditions such as ischemic heart disease, chronic wounds, and neurovascular disorders. Benzoquinolines, for instance, could simultaneously promote new vessel formation and enhance blood perfusion in ischemic tissues, potentially improving therapeutic outcomes in cardiovascular diseases, wound healing, stroke recovery, and cancer therapy by facilitating better delivery of therapeutic agents [24,25,26,27,28].
Artemia salina (brine shrimp) serves as a valuable model organism for vascular biology research due to its transparency, rapid development, and ethical advantages. Artemia larvae enable direct, non-invasive visualization of vascular development and quickly form vascular networks in response to angiogenic stimuli such as vascular endothelial growth factor (VEGF) and fibroblast growth factors (FGFs), thus effectively mimicking aspects of human angiogenesis [29,30,31].
The use of Artemia salina provides key advantages, including transparency, rapid developmental timelines, cost-effectiveness, ethical acceptability compared to mammalian models, and suitability for high-throughput screening. Insights derived from Artemia-based assays effectively support subsequent investigations in complex mammalian models and ultimately clinical research [32,33,34,35].
In summary, this study utilizes the Artemia salina model to investigate angiogenic and vasodilatory effects of quinoline derivatives and prostaglandin E1 (PGE1), providing an efficient, ethical, and informative framework to evaluate therapeutic candidates for enhancing vascular growth and circulation.
Comment5: Summarize the introduction, and maybe a graphical abstract or flow chart can help to clarify the large number of models and items introduced.
Response 5:Two flow charts were added to the materials and methods section, which was completely rewritten.
Comment 6: Please rewrite materials and methods, this section needs to be a clear guide to (eventually) reproduce the experiment and not only a list of what you have used. Please revise this section, clarifing the steps and the aim of each process.
Response 6: the following text was added to replace material and mehods:
Materials and Methods
- Materials and Methods
2.1. Model Organism: Artemia salina
Artemia salina, also known as brine shrimp, is a microcrustacean widely utilized in developmental biology and pharmacological testing due to its optical transparency, rapid embryogenesis, and ethical acceptability for high-throughput screening. Adult Artemia typically measure 8–15 mm in length and feature a segmented body divided into head, thorax, and abdomen. The thorax contains 11 pairs of phyllopodia, which facilitate locomotion, feeding, and respiration. The transparent exoskeleton and hemolymph circulation enable direct visualization of vascular-like structures, making Artemia an ideal model for studying angiogenesis and vasculogenesis.
2.2. Preparation and Hatching of Artemia salina Cysts
2.2.1. Hydration and Decapsulation
Dry cysts of Artemia salina (INVE Aquaculture) were hydrated by immersing them in sterile artificial seawater (ASW; 35 ppt salinity, pH ~8.0) for 1 hour at room temperature (22–25°C). Hydration activates metabolic pathways in the embryos. Following hydration, cysts were decapsulated by treatment with 1% sodium hypochlorite (NaOCl) for precisely 10 minutes under constant, gentle agitation. This step removes the hard chorionic layer, which improves hatching rates and reduces the risk of contamination. The decapsulation was terminated by transferring the cysts into sterile distilled water, followed by five sequential rinses to eliminate residual hypochlorite and ensure safe hatching conditions.
2.2.2. Hatching Conditions
The rinsed, decapsulated cysts were incubated in glass vessels filled with fresh ASW under standardized conditions: 28 ± 1 °C, continuous illumination (2000–3000 lux), and gentle aeration using sterile air stones. Nauplii typically hatch within 20–24 hours. Only first-instar nauplii exhibiting normal morphology and active swimming behavior were selected for experimentation.
2.3. Test Molecule Treatments
2.3.1. Experimental Grouping and Exposure Design
Selected nauplii were transferred into 12-well sterile culture plates, with 10–15 nauplii per well and 3 mL of test solution. Each condition was replicated in triplicate (n = 3). Control groups were maintained in ASW alone. Nauplii were exposed to test compounds for 72 hours under the same conditions as described above, and imaging was conducted at 24, 48, and 72 hours.
2.3.2. PGE₁ (Prostaglandin E1) Treatment
PGE₁ (Sigma-Aldrich, ≥98% purity) was dissolved in DMSO to produce a 10 mM stock solution, then serially diluted in ASW to final concentrations of 0.01, 0.1, 1, and 10 µM. Final DMSO concentration did not exceed 0.1%. Based on pilot studies, one µM was identified as the optimal concentration for pro-angiogenic effects and used in subsequent assays.
2.3.3. Quinoline Derivative Treatments
Five quinoline-based compounds were tested: 3-aminoquinoline, 5-aminoquinoline, 8-aminoquinoline, 8-hydroxyquinoline, and unsubstituted quinoline (all ≥98% purity, Sigma-Aldrich). Stock solutions (10 mM in DMSO) were diluted to working concentrations (1–100 µM). Final effective concentrations were 25 µM for aminoquinolines and 75 µM for 8-hydroxyquinoline and quinoline, determined via preliminary toxicity and activity profiling(Table 1).
2.4. Morphological and Vascular Imaging
Nauplii were immobilized with 1% methylcellulose on clean microscope slides and covered with a coverslip. Images were acquired using a stereomicroscope (Leica M80) equipped with a digital camera (Leica MC170 HD) at magnifications ranging from 20x to 80x. Imaging was conducted at 24, 48, and 72 hours post-exposure to document developmental progression and vascular growth.
2.4.1. Quantitative Vascular Analysis Using AngioTool
Captured images were analyzed using AngioTool v0.6, an image processing software for quantifying angiogenesis parameters[36]. The steps were as follows: (1)Input Image Preparation: Bright-field images were saved as high-resolution TIFF or PNG files. (2)Launch AngioTool v0.6: Open the software and load each image using the "Open Image" function.(3) Set Thresholds: Use the automatic or manual threshold tool to highlight the vascular structures. Adjust “Min/Max Vessel Diameter” and “Intensity Threshold” for optimal contrast.(4) Run Analysis: Click “Analyze” to process the image. The software outputs the following parameters: Total vessel area, Vessel percentage coverage, Total number of junctions, Junction density, Average and total vessel length, Number of endpoints, Vascular symmetry (lateralization index)(5). Export Data: Save results and overlay images using the export function for each sample. The same settings were applied across all experimental images to maintain consistency. Visual outputs were verified for segmentation accuracy, and erroneous traces were manually corrected where needed.
2.5. Graph-Theoretical Analysis of Vascular Networks
To assess vascular architecture beyond standard morphometrics, digital images were converted into undirected weighted graphs for topological and complexity analysis. The process was carried out using Python (v3.9) and the NetworkX library, as outlined below[37,38]:
Image-to-Graph Conversion Workflow:
- Image Preprocessing: Convert vascular images to binary (black & white) using FIJI or Python's OpenCV.Skeletonize the image to reduce vessel structures to single-pixel-wide lines.
- Node and Edge Extraction: Nodes were identified at each bifurcation or terminal endpoint. Edges were drawn between nodes along the skeleton lines.
- Graph Construction (Python + NetworkX): Import the skeletonized image as a pixel matrix.Use NetworkX to create a Graph() object:
import networkx as nx
G = nx.Graph()
G.add_nodes_from(node_list)
G.add_edges_from(edge_list)
Calculate edge weights by measuring the Euclidean distance between connected nodes.
- Graph Metrics Computation: Using NetworkX functions, the following descriptors were extracted:number_of_nodes(G); number_of_edges(G); nx.average_degree_connectivity(G;nx.clustering(G; nx.betweenness_centrality(G);nx.density(G). Custom modularity and fractal dimension scripts were used to complete network profiling.
- Data Export and Visualization: Graph data were exported as CSV for statistical analysis. Topological maps, radar plots, and combo plots were generated using Matplotlib and Seaborn. This graph-theoretic approach enabled detailed analysis of vascular complexity, including connectivity, redundancy, and local vs. global organization across treatment groups. The study was carried out in three main directions: (a) Generation of good resolution images in order to ensure proper image conversion and analysis; (b) Representation of Vascular Networks: Graph model—vascular networks are represented as graphs where nodes (or vertices) represent junction points or branching points of blood vessels, and edges represent the blood vessels connecting these junctions; (c) Types of graphs generated were undirected graphs - standard for analyzing connectivity and topology; weighted graphs- edge weights can represent vessel diameter.
Once the graph was extracted, key metrics such as the number of nodes, number of edges, average degree, average clustering coefficient, and average betweenness centrality were computed. The data were calculated using the online computational engine Wolfram Alpha [39,40,41,42,] and graphically represented using radar and combo plots.
2.6. Statistical Analysis
All data are presented as mean ± standard deviation (SD). Statistical differences between groups were analyzed using one-way ANOVA, followed by Tukey’s multiple comparison test for pairwise comparisons. For non-parametric data distributions, Kruskal–Wallis tests with Dunn’s post hoc analysis were applied. Statistical significance was defined as p < 0.05. All analyses were conducted using GraphPad Prism v9.0 and SPSS v26. Results were visualized with box plots and bar graphs to illustrate group differences, variability, and confidence intervals[43,44].
Overall, Figures 1 and Figure 2 depict a flow chart representing the methodology of the study and the entire study design, respectively.
Figure1.. A flow chart of the study methodology is shown. (1) Preparation of Artemia salina Cysts - The experimental process began with the preparation of Artemia salina cysts. These dormant cysts were stored under dry conditions and reactivated through a hydration process using artificial seawater (ASW). This rehydration marked the initiation of metabolic processes critical for embryonic development.(2)Hydration (1 hour in ASW)-Cysts were immersed in ASW for one hour at room temperature. This step served to reactivate the embryonic metabolic machinery, initiating physiological development and rendering the cysts responsive to further manipulation.(3)-Decapsulation (1% Hypochlorite, 10 min)-following hydration, the cysts were subjected to decapsulation by immersion in a freshly prepared 1% sodium hypochlorite solution for precisely 10 minutes. This chemical treatment removed the protective chorion layer, increasing hatching efficiency and promoting synchronized development across experimental replicates.(4) Washing Cysts (Distilled Water Rinsing) - After decapsulation, the cysts were carefully rinsed multiple times with distilled water to eliminate residual hypochlorite. This was essential to prevent chemical toxicity and ensure high viability during subsequent hatching.(5)Hatching Procedure: Decapsulated and washed cysts were incubated in ASW at 28°C under constant illumination, with continuous aeration. These conditions facilitated efficient oxygenation and supported uniform hatching of nauplii, which typically emerged within 20–24 hours.(6) Selection of Uniform Nauplii (24 hours post-hatching)-At 24 hours post-hatching, nauplii exhibiting normal morphology and active swimming behavior were selected. Uniform selection ensured consistency across control and treatment groups and reduced biological variability in downstream analyses.(7)Experimental Treatments- Selected nauplii were divided into experimental and control groups. The control group was maintained in plain artificial seawater (ASW). In contrast, the experimental groups were treated with two classes of compounds: PGE₁. Treatments: Nauplii were exposed to Prostaglandin E₁ at concentrations of 0.01 µM, 0.1 µM, 1 µM, and 10 µM. The optimal working dose, identified through pilot assays, was 1 µM. Quinoline Derivative Treatments: Separate groups were treated with aminoquinolines (3-, 5-, and 8-aminoquinoline) at 25 µM, and 8-hydroxyquinoline or unsubstituted quinoline at 75 µM.(8)Morphological & Vascular Assessment- Morphological development and vascular structure formation were assessed using stereomicroscopy at 24, 48, and 72 hours post-treatment. High-resolution images were captured to document appendage formation, body growth, and vascular branching, enabling qualitative comparisons across treatment groups.(9) Quantitative Vascular Analysis-Captured images were analyzed using AngioTool v0.6, a specialized software for vascular network quantification. Parameters measured included total vessel area, percentage vessel coverage, total and average vessel lengths, number of junctions, junction density, endpoints, and laterality. These quantitative metrics provided objective comparisons of angiogenic effects elicited by each compound.(10) Graph-Theoretical and Statistical Analysis-To further analyze the structural characteristics of the vascular networks, digital images were converted into graph models where vessel junctions were represented as nodes and vessels as edges. Graph metrics such as node and edge count, average node degree, clustering coefficient, and betweenness centrality were computed. These were used to evaluate vascular complexity and connectivity.Finally, all quantitative data were subjected to statistical analysis using ANOVA to identify significant differences between control and treatment groups. Statistical significance was defined at p < 0.05, validating the biological relevance of the observed angiogenic responses.
Figure 2. Flow chart of the study. This flow chart illustrates a stepwise approach to understanding and analyzing endothelial network formation, focusing on both biological mechanisms and experimental methodologies. The process can be broadly divided into experimental modeling, dynamic observation, mechanistic understanding, and physiological validation.(1)Development of In Vitro Endothelial Cell Network Models. This foundational step involves creating laboratory models to mimic endothelial networks. These models are essential for systematically studying cellular behavior in a controlled environment. (2) Image-based Quantification: High-resolution imaging techniques to quantify network properties. This allows researchers to measure parameters such as branching, node formation, and connectivity. (3)Controlled Assay Development- Assays are optimized for consistent and repeatable measurements. The aim is to establish clear conditions that allow for better observation and analysis of network formation and dynamics.(4) Time-lapse Imaging and Quantitative Analysis- Using time-lapse microscopy, researchers can visualize the dynamics of network formation. This provides insight into how endothelial cells migrate, connect, and evolve. (5) Mechanistic studies involve dissecting the feedback mechanisms and remodeling processes that influence network formation. By understanding these mechanisms, researchers can identify potential targets for therapeutic intervention. (6) Candidate Screening and Validation: Potential preliminary tests to identify and validate potential factors affecting network formation. This step helps narrow down the key players involved in network dynamics. (7) Physiological Relevance - Finally, validated mechanisms and factors are tested in physiologically relevant contexts to assess their in vivo significance. This ensures that the findings translate into meaningful biological insights.
Comment 7: Results: In this form I'm not capable to follow the description of the results and the meaning of the work, I suggest to improve the value of this work dividing in subsection the results, ghiving to this subsection a short title that summarize the main data (as you can see in other articles from this journal). The bargraphs have to be clear, if needed devide the data to optimize the visualization of the trand (for example in graph 7-8 the orage bar is too high and you can lose the modulation in the blue one). Statistical analysis is needed and has to be reported or described.
Response 7: The result of the action was organized with headings and corrected as the reviewer suggested. References were also added to support the paragraphs. Also, a statistical part heading and paragraphs were added.
Commet 8: the English is in a strange, chaotic form, and there are numerous typos
Response 8 : Thank you very much for highlighting the linguistic concerns. We fully acknowledge the importance of clear, accurate, and professional language in scientific communication. To address your feedback thoroughly, the manuscript has been carefully reviewed and revised by a native English-speaking editor. This comprehensive linguistic revision ensures clarity, readability, and coherence throughout the manuscript. We have corrected all typographical errors, improved sentence structure, and standardized terminology to eliminate confusion and enhance overall readability. We believe these improvements significantly improve the clarity and scientific rigor of the manuscript, facilitating a better understanding of our research findings.
All corrections were performed in red.
Round 2
Reviewer 2 Report
Comments and Suggestions for Authors
The manuscript has been extensively revised, improving clarity and coherence. I would like to thank the authors for addressing my suggestions.